# Light availability modulates the effects of warming in a marine N$_2$ fixer

Xiangqi Yi[1], Fei–Xue Fu[2], David A. Hutchins[2], Kunshan Gao[1,3]

[1]State Key Laboratory of Marine Environmental Science, College of Ocean and Earth Sciences, Xiamen University, Xiamen, China
[2]Department of Biological Sciences, University of Southern California, Los Angeles, CA, USA
[3]Co-Innovation Center of Jiangsu Marine Bio-industry Technology, Jiangsu Ocean University, Lianyungang 222005, China

*Correspondence to*: Kunshan Gao (ksgao@xmu.edu.cn)

**Abstract.** *Trichodesmium* species, as a group of photosynthetic N$_2$ fixers (diazotrophs), play an important role in the marine biogeochemical cycles of nitrogen and carbon, especially in oligotrophic waters. How ongoing ocean warming may interact with light availability to affect *Trichodesmium* is not yet clear. We grew *Trichodesmium erythraeum* IMS 101 at three temperature levels of 23, 27 and 31 °C under growth limiting and saturating light levels of 50 and 160 µmol quanta m$^{-2}$ s$^{-1}$, respectively, for at least 10 generations, and then measured physiological performances, including specific growth rate, N$_2$ fixation rate and photosynthesis. Light availability significantly modulated the growth response of *Trichodesmium* to temperature, with the specific growth rate peaking at ~27 °C under the light–saturating conditions, while growth of light–limited cultures was non–responsive across the tested temperatures (23, 27 and 31 °C). Short-term thermal responses for N$_2$ fixation indicated that both high growth temperature and light intensity increased the optimum temperature (T$_{opt}$) for N$_2$ fixation and decreased its susceptibility to supra–optimal temperatures (deactivation energy, E$_h$). Simultaneously, all light–limited cultures with low T$_{opt}$ and high E$_h$ were unable to sustain N$_2$ fixation during short–term exposure to high temperatures (33–34 °C) that are not lethal for the cells grown under light–saturating conditions. Our results imply that *Trichodesmium* spp. growing under low light levels while distributed deep in the euphotic zone or under cloudy weather conditions might be less sensitive to long-term temperature changes that occur on the time scale of multi-generation but more susceptible to abrupt (less than one generation time span) temperature changes, such as those induced by cyclone and heat waves.

## 1. Introduction

In vast areas of the oceans, primary production is usually limited by availability of nitrogen (Moore et al., 2013). In addition to recycling within the euphotic zone, biologically available nitrogen sources can be supplied to phytoplankton from upwelling, aerosol deposition and $N_2$ fixation by diazotrophic prokaryotes, supporting new primary production (Dugdale and Goering, 1967). *Trichodesmium* is one of the major diazotrophic organisms occurring in the pelagic oceans (Zehr, 2011). *Trichodesmium* is a genus of filamentous cyanobacteria that exists as both single filaments and colonies consisting of tens to hundreds of trichomes, and that is broadly distributed in oligotrophic tropical and subtropical oceans (Capone et al., 1997). Its contribution to local new production can even be more important than that of nitrate diffusion in some regions (Capone et al., 2005; LaRoche and Breitbarth, 2005; Mahaffey et al., 2005), and it thus plays a significant role in marine ecosystems and biogeochemical cycles of nitrogen and carbon (Sohm et al., 2011; Zehr, 2011).

*Trichodesmium* has attracted tremendous research interest since its discovery as diazotroph in the early 1960s (Dugdale et al., 1961; Dugdale et al., 1964). Recently, considerable research attention has been focused on evaluating effects of the ongoing ocean climate changes, including sea surface warming associated with global warming, on this keystone organism (Fu et al., 2014; Hutchins and Fu, 2017; Jiang et al., 2018). The IPCC RCP 8.5 scenario predicts that upper ocean temperature will increase by about 3 ℃ on average by the end of the 21st century, and the strongest ocean warming will happen in tropical and subtropical regions (Collins et al., 2013). Because of its important role in marine biogeochemical cycles and marine ecosystems, understanding the responses of *Trichodesmium* to ocean warming and their underlying mechanisms will be critical to evaluating the potential implications of climate changes on marine primary productivity, food web dynamics and biogeochemical cycles.

Previous studies demonstrate that without resource limitation the growth versus temperature curve is unimodal in *Trichodesmium* with lower and upper tolerance limits separately at 18-20 ℃ and 32-34 ℃ and optimum temperature at 26-28 ℃ (Breitbarth et al., 2007; Chappell and Webb, 2010; Fu et al., 2014). Based on these findings and the spatial heterogeneity of present temperatures and projected warming of *Trichodesmium's* habitat (Capone et al., 1997; Collins et al., 2013), the effects of ocean warming on *Trichodesmium* can be spatially diverse, generally benefiting those occurring in relatively high latitude but being harmful to those occurring near to the equator (Breitbarth et al., 2007; Fu et al., 2014

Thomas et al., 2012). However, this pattern can be distorted and complicated by resource limitations. For example, it is shown that iron limitation, which is commonly experienced by *Trichodesmium* in nature (Hutchins and Boyd, 2016; Sohm et al., 2011), substantially increases the optimum temperature in *Trichodesmium* (Jiang et al., 2018).

Similar to iron availability, light is also among the key environmental drivers for *Trichodesmium* (Cai and Gao, 2015; Cai et al., 2015; Breitbarth et al., 2008). *Trichodesmium* can be distributed from the sea surface down to 150 m depth where light intensity at noon ranges from > 2000 µmol quanta $m^{-2}$ $s^{-1}$ to < 10 µmol quanta $m^{-2}$ $s^{-1}$ (Davis and McGillicuddy, 2006; Olson et al., 2015). Moreover, *Trichodesmium* is known to be able to partially regulate its vertical position in water column by buoyancy adjustment (Villareal et al. 2003). Currently, ocean warming effects on *Trichodesmium* have been widely examined under single, saturating light conditions, providing important knowledge on this diazotroph's physiological responses to temperature changes (Breitbarth et al., 2007; Fu et al., 2014; Jiang et al., 2018; Levitan et al., 2010). However, how it responds to warming under both light–limiting and saturating conditions is also of general significance, but has been rarely studied (Boatman et al., 2017).

The growth of phytoplankton is a holistic result of many biochemical and physiological activities, so its responses to environmental changes are dependent on species-specific physiology. Normally, warming increases enzyme activities, accelerating biochemical reactions (Gillooly et al., 2001). For phytoplankton, reduced growth at high temperature might be the result of less carbon availability due to the higher thermal sensitivity of respiration compared to that of photosynthesis (Padfield et al., 2015). This negative effect of high temperature might be exacerbated by directly reduction of photosynthesis under light limitation. In the diazotroph *Trichodesmium*, besides photosynthesis and respiration, $N_2$ fixation process might also play a critical role in its growth response to environmental changes. Little has been documented on this aspect (Jiang et al., 2018).

In this study, we explored the combined effects of temperature and light in *Trichodesmium erythraeum* IMS 101. We measured the specific growth rate, photosystem functions and $N_2$ fixation rate in *Trichodesmium* cultures acclimated to two light levels and three temperatures. Moreover, we measured their short-term thermal responses for $N_2$ fixation. In this paper, "acclimation" and "acclimated" indicate that cultures were given enough time (several weeks) to respond to the environmental changes so that balanced growth was achieved, and "short-term" refers to acute processes and changes

occurring within hours. Although adaptation (over several hundreds of generations) has been demonstrated to be critical in evaluating the responses in phytoplankton to environmental changes (Hutchins et al., 2015; Li et al., 2017; Padfield et al., 2015; Schaum et al., 2018; Tong et al., 2018), it is beyond the scope of this study.

## 2. Material and methods

### 2.1 Culture conditions

Triplicate cultures of *Trichodesmium erythraeum* (strain IMS101, originally isolated from the North Atlantic Ocean by (Prufert–Bebout et al. 1993)) were established under six different culture conditions. These included factorial combinations of three temperatures (23±1, 27±1 and 31±1 °C) and two light intensities (saturating light, $160 \pm 20$ µmol quanta $m^{-2}$ $s^{-1}$ and limiting light, $50 \pm 6$ µmol quanta $m^{-2}$ $s^{-1}$). These three growth temperatures are representatives of present and future temperatures of *Trichodesmium* habitats (Breitbarth et al., 2007; Fu et al., 2014). The limiting and saturating light levels were established based on a pilot experiment (Supplementary Fig. S1(a)) and previous studies (Cai et al., 2015; Breitbarth et al., 2008; Garcia et al., 2011). All the cultures were run semi–continuously by continual dilutions (every 2–4 days) in the artificial seawater medium YBCII without combined nitrogen (Chen et al., 1996) in 1–L glass flasks maintained in plant growth chambers (HP300G–C, Ruihua, China). Light was provided by LED tubes (FSL, China) with a 12:12 Light:Dark cycle. Different levels of light intensity were achieved using neutral density filters. Cultures were continuously bubbled with air (outdoor) so that the cyanobacteria floated as single filaments. The cells were allowed to acclimate to each condition for at least 10 generations. Acclimation was confirmed by balanced growth with stable specific growth rate. Then, we started the sampling and data collection.

### 2.2 Chlorophyll–a (Chl–a) concentration and specific growth rate

Chl–a concentration was spectrophotometrically quantified by gently filtering the cells onto glass–fiber filters (GF/F, Whatman), followed by extraction in pure methanol at 4 °C for 24 h and centrifugation at 6000g for 10 minutes. The absorbance spectrum of the supernatant was determined from 400 nm to 700 nm using a spectrophotometer (DU800, Beckman, USA). Chl–a concentration was calculated as: [Chl–a] (µg $mL^{-1}$) $=12.9447*(A_{665}–A_{750})$, where $A_{665}$ and $A_{750}$ were respectively the absorbances at 665 and 750 nm (Ritchie, 2006).

100    Because Chl-a concentration is a good proxy for biomass in *Trichodesmium* (Breitbarth et al. 2007), Chl-a concentrations

measured at different days were analysed using Eq. (1) to obtain the specific growth rate:

$$\ln(\text{Chl-a }(t)) = \mu * t + b \tag{1}$$

where Chl-a(t) is the Chl-a concentration at time t (d), $\mu$ is the specific growth rate ($d^{-1}$), and b is interpreted as the natural

logarithm of Chl-a concentration at day 0. Because the cultures were semi–continuously maintained, Chl–a concentrations at

105    each time point was corrected by the dilution ratios with the assumption of no dilutions.

## 2.3 Short–term thermal response for $N_2$ fixation

$N_2$ fixation rates were determined using the acetylene reduction assay assuming a ratio of 4:1 to convert ethylene production

to $N_2$ fixation (Capone, 1993). To examine the responses of $N_2$ fixation in the cells grown at different temperatures and light

levels to short-term temperature changes, we simultaneously measured $N_2$ fixation at five temperatures ranging from 19 to

110    35 °C. For each replicate, a 25ml aliquot of the culture was taken and dispensed into five vials. Five vials each of which

contained 5ml culture were separately placed in five different zones of two multi–zone culture chambers (HP100–2 and

HP100–3, Ruihua, China) and allowed to equilibrate to different target temperatures for 30 minutes. Pilot experiments had

showed that 30 minutes was enough for temperature equilibrium. After temperature equilibration, each vial was spiked with

1 ml (12.5% of headspace volume) pure acetylene and incubated for another 30 minutes under the growth light level. The

115    quantity of ethylene produced was determined using a gas chromatograph with flame ionization detector (Clarus 580,

PerkinElmer, USA).

Typically, the short-term thermal response curves for $N_2$ fixation were unimodal and negatively skewed, which could be

accommodated to a modified version (Padfield et al., 2015; Schaume et al. 2018) of the Sharpe–Schoolfield model

(Schoolfield et al., 1981; Sharpe and DeMichele, 1977):

$$N(T) = N(T_c) * \exp\left(E_a * \left(\frac{1}{kT_c} - \frac{1}{kT}\right)\right) / \left(1 + \exp\left(E_h\left(\frac{1}{kT_h} - \frac{1}{kT}\right)\right)\right) \tag{2}$$

where N(T) is the $N_2$ fixation rate ($\mu$mol $N_2$ mg Chl–$a^{-1}$ $h^{-1}$) at temperature T (Kelvin, K), $E_a$ is the activation energy

(electron volt, eV) for $N_2$ fixation, being indicative of the steepness of the slope leading to a thermal optimum, $E_h$ is the

deactivation energy (electron volt, eV) characterizing high temperature induced inactivation above deactivation temperature

$T_h$ (K), and $N(T_c)$ is the $N_2$ fixation rate at an arbitrary reference temperature $T_c$ (here, $T_c$ = 25 °C) used for normalization.

According to Eq. (2), the optimum temperature ($T_{opt}$, K) corresponding to the maximal N$_2$ fixation rate ($N_{max}$, µmol N$_2$ mg Chl–a$^{-1}$ h$^{-1}$) is:

$$T_{opt} = E_h * T_h / (E_h + k * T_h * \ln(E_h/E_a))$$  (3)

Additionally, we also obtained the N$_2$ fixation rate at growth temperature ($N_{growth}$) by bring corresponding growth temperatures to Eq. (2).

## 2.4 Chl–a fluorometry

Photosystem II (PSII) effective quantum yield ($\Phi_{PSII}$) and photosynthetic relative electron transport rate (rETR) were measured by using the Multiple Excitation Wavelength Chlorophyll Fluorescence Analyzer (MULTI–COLOR–PAM, Walz, German) equipped with the US–T temperature control unit (Walz, Germany). Aliquots of 1.5 ml of the culture were taken to determine effective quantum yield ($\Phi_{PSII}$) under actinic light levels that were the same as those of the growth conditions. Then, $\Phi_{PSII}(E)$ values were successively measured at seven levels of light intensity (E) ranging from 0 to 1064 µmol quanta m$^{-2}$ s$^{-1}$. Samples were allowed to acclimate to each light level for 3 minutes before $\Phi_{PSII}(E)$ measurements (Suggett et a., 2007). Relative electron transport rate (rETR) at each light level was calculated as: rETR = E * $\Phi_{PSII}(E)$ (Ralph and Gademann, 2005). The light response curve of rETR was analysed according to the model of (Eilers and Peeters, 1988):

$$rETR = \frac{E}{a*E^2 + b*E + c}$$  (4)

Photosynthetic parameters including photosynthetic light harvesting efficiency (α), rETR maximum (rETR$_{max}$) and light saturation point (E$_k$) can be calculated as:

$$\alpha = \frac{1}{c}$$  (5)

$$rETR_{max} = \frac{1}{b + 2*\sqrt{a*c}}$$  (6)

$$E_k = \frac{c}{b + 2*\sqrt{a*c}}$$  (7)

During the measurements, sample temperature was maintained at the corresponding growth temperatures using the US–T temperature control unit (Walz, Germany).

**2.5 Statistical analyses**

Statistical analyses were performed with the R language (version 3.5.3). [Chl-a] versus time in each of the triplicate cultures for each treatment was fitted to Eq. (1) using function "lm" in package "stats" to get the specific growth rate ($\mu$). The significance of differences in specific growth rate ($\mu$) between treatments was tested using two-way analyses of variance (ANOVA) (function "aov" in package "stats"), followed by Tukey's test for pairwise comparison (function "TukeyHSD" in package 'stats'). The homogeneity of variance assumption and the residuals normality assumption were separately checked by Levene's test (function "leveneTest" in package "stats") and Shapiro-Wilk test (function "shapiro.test" in package "stats"). The significance level was set to 0.05.

To test whether parameters $N(T_c)$, $E_a$, $E_h$, $T_h$ and $T_{opt}$ differ between different treatments, we fitted short-term thermal responses for $N_2$ fixation to Eq.(2) using nonlinear mixed effects models ("nlme" package). Models included random effects on each of the parameters of Eq.(2) by replicate. The structure of the fixed effects of the nonlinear mixed effects model was determined by trying all possible models (625 models) and selecting the one with the lowest small sample-size corrected Akaike information criterion (AICc) (see Supplementary Table S3 for all tested models' AICc). AICc was calculated using function "AICc" in package "MuMIn".

Light curve of rETR in each of the triplicate cultures for each treatment was fitted to Eq. (4) using function "nls" in package "stats". The significance of differences in photosynthetic light harvesting efficiency ($\alpha$), rETR maximum ($rETR_{max}$), light saturation point ($E_k$) and effective quantum yield ($\Phi_{PSII}$) between treatments was tested using the same statistical methods as those for $\mu$. The significance level was set to 0.05.

**3. Results**

**3.1 Specific growth rate and $N_2$ fixation rate**

Specific growth rates of *Trichodesmium* IMS 101 were significantly affected by growth light intensity (two-way ANOVA, $F_{1,12} = 662.7$, $P < 0.001$), growth temperature (two-way ANOVA, $F_{2,12} = 22.0$, $P < 0.001$) and the interaction between these two drivers (two-way ANOVA, $F_{2,12} = 18.0$, $P < 0.001$) (Table 1). High growth light intensity increased specific growth rates of *Trichodesmium* IMS 101 by 63% at 23 °C (Tukey's test comparing light-saturated and light-limited growth rates at 23ºC,

$P < 0.001$), 111% at 27 °C (Tukey's test comparing light-saturated and light-limited growth rates at 27 °C, $P < 0.001$) and

88% at 31 °C (Tukey's test comparing light-saturated and light-limited growth rates 31 °C, $P < 0.001$), respectively (Fig.

1(a)). The interaction between growth light intensity and temperature on specific growth rate was indicated by the totally

different temperature effects between light-saturated and light-limited cultures. Light–saturated growth rates of

*Trichodesmium* IMS 101 were maximal at 27 °C with a value of $0.52 \pm 0.02$ d$^{-1}$ ($\pm$SD), being higher by 29.5% (Tukey's test

comparing growth rates between light-saturated cultures grown at 27 °C and 23 °C, $P < 0.001$) and 21.3% (Tukey's test

comparing growth rates between light-saturated cultures grown at 27 °C and 31 °C, $P < 0.001$) than those at 23 °C and 31 °C,

respectively. However, light–limited growth rates ranged from $0.23 \pm 0.02$ d$^{-1}$ ($\pm$SD) to $0.25 \pm 0.01$ d$^{-1}$ ($\pm$SD), with no

significant differences between the tested temperatures (Tukey's test comparing growth rates among light-limited cultures

grown at three temperatures, $P > 0.05$ for all three comparisons).

Overall, N$_2$ fixation rates at growth temperature (N$_{growth}$) (Fig. 1(b)) were significantly higher in cultures grown under high

light intensity compared to those grown under low light intensity (two-way ANOVA, $F_{1,12} = 149.9$, $P < 0.001$). Also, growth

temperature significantly affected N$_{growth}$ (two-way ANOVA, $F_{1,12} = 3912.3$, $P < 0.001$). Different thermal effects between

light-saturated and light-limited cultures indicated a significant interaction between light and temperature on N$_{growth}$ (two-

way ANOVA, $F_{2,12} = 112.7$, $P < 0.001$). For light-saturated cultures, the N$_{growth}$ peaked at 27 °C with a value of $17.1 \pm 0.5$

µmol N$_2$ mg Chl–a$^{-1}$ h$^{-1}$ ($\pm$SD),  which was higher by 39% and 17%  than those at 23 °C (Tukey's test comparing N$_{growth}$

between light-saturated cultures grown at 27 °C and 23 °C, $P < 0.001$) and 31 °C (Tukey's test comparing N$_{growth}$ between

light-saturated cultures grown at 27 °C and 23 °C, $P < 0.001$), respectively. However, for light-limited cultures, the value of

N$_{growth}$ at 27 °C ($6.8 \pm 0.2$ µmol N$_2$ mg Chl–a$^{-1}$ h$^{-1}$ ($\pm$SD)) was similar to that at 23 °C ($6.3 \pm 0.1$ µmol N$_2$ mg Chl–a$^{-1}$ h$^{-1}$ ($\pm$SD))

(Tukey's test comparing N$_{growth}$ between light-limited cultures grown at 27 °C and 23 °C, $P = 0.54$) but significantly higher

than that at 31 °C ($3.8 \pm 0.4$ µmol N$_2$ mg Chl–a$^{-1}$ h$^{-1}$ ($\pm$SD)) (Tukey's test comparing N$_{growth}$ between light-limited cultures

grown at 27 °C and 31 °C, $P < 0.001$).

### 3.2 PSII effective quantum yield ($\Phi_{PSII}$) and rETR light response curves

Compared to light–saturated cells, light–limited cells had higher values of $\Phi_{PSII}$ (Fig. 1(c); two-way ANOVA, $F_{1,12} = 233.2$,

$P < 0.001$). Meanwhile, under both light regimes, $\Phi_{PSII}$ in *Trichodesmium* cultures grown at 31 °C was significantly higher

than that in cultures grown at 23 °C and 27 °C (two-way ANOVA, $F_{2,12} = 22.1$, $P < 0.001$). No interaction between growth light intensity and temperature on $\Phi_{PSII}$ was found (two-way ANOVA, $F_{2,12} = 1.8$, $P = 0.211$).

The rETR light response curve of *Trichodesmium* IMS 101 was influenced by growth temperature in both light–saturated (Fig. 2(a)) and light–limited (Fig. 2(b)) treatments. This thermal impact was mainly reflected in the $rETR_{max}$ (two-way

ANOVA, $F_{2,12} = 31.2$, $P < 0.001$), which tended to be higher in cultures acclimated to 31 °C than those in cultures acclimated to 23 °C or 27 °C (Table 2; Fig. 2). Additionally, high growth light intensity tended to decreased the $rETR_{max}$ (two-way ANOVA, $F_{1,12} = 31.2$, $P < 0.001$), especially for cultures grown at 27 °C (Tukey's test comparing $rETR_{max}$ between light-saturated and light-limited cultures grown at 27 °C, $P < 0.001$). Both high light intensity (two-way ANOVA, $F_{1,12} = 6.0$, $P < 0.05$) and high temperature (two-way ANOVA, $F_{2,12} = 5.0$, $P < 0.05$) significantly increased the $E_k$ (Table 2).

**3.3 Short–term thermal response for $N_2$ fixation**

Optimum temperature ($T_{opt}$) for $N_2$ fixation in *Trichodesmium* IMS 101 was affected by both growth temperature and light intensity (Table 3). Generally, $T_{opt}$ for $N_2$ fixation in light-saturated cultures were higher than that in light-limited cultures, whereas warming effects on $N_2$ fixation $T_{opt}$ differed between light-saturated and light-limited cultures. For light-saturated cultures, elevations of growth temperature raised the $T_{opt}$ for $N_2$ fixation. A 4 °C warming was accompanied by 0.5-0.8 °C

increase of $T_{opt}$, which was 28.7 ± 0.2 °C (±S.E.M), 29.5 ± 0.2 °C (±S.E.M) and 30.0 ± 0.3 °C (±S.E.M)) for the cells grown at at 23 °C, 27 °C and 31 °C,  respectively. Under limiting light level, $T_{opt}$ in cultures grown at 27 °C ($T_{opt} = 28.6 ± 0.2$ °C (±S.E.M)) was higher than that in cultures grown at 23 °C ($T_{opt} = 28.2 ± 0.2$ °C (±S.E.M)), but $T_{opt}$ in cultures grown at 31 °C ($T_{opt} = 27.8 ± 0.2$ °C (±S.E.M)) was the lowest among all treatments. As expected, the maximal $N_2$ fixation rate ($N_{max}$) in light-saturated cultures was higher than that in light-limited culture (Table 3). The temperature effect on $N_{max}$ was also

dependent on the light availability. Light-saturated $N_{max}$ was highest in cultures grown at 27 °C ($N_{max} = 19.3 ± 0.4$ μmol $N_2$ mg Chl–a $^{-1}$ h$^{-1}$ ( ±S.E.M)), being higher by 21% and 32% than those grown at 23 °C ($N_{max} = 16.0 ± 0.3$ μmol $N_2$ mg Chl–a $^{-1}$ h$^{-1}$,( ±S.E.M)) and 31 °C ($N_{max} = 14.6 ± 0.4$ μmol $N_2$ mg Chl–a $^{-1}$ h$^{-1}$,( ±S.E.M)), respectively. However, $N_{max}$ for light-limited cultures was similar among different temperature treatments (Table 3).

The value of deactivation energy ($E_h$) for $N_2$ fixation, reflecting the thermal susceptibility to supra–optimal temperatures,

was affected by both light availability and growth temperature, but not by their interaction (Table 3). $E_h$ tended to be lower in

*Trichodesmium* cultures grown under high temperature and high light intensity. With the highest $T_{opt}$ (30.0 ± 0.3 °C (±S.E.M)) and the lowest $E_h$ (1.47 ± 0.14 eV(±S.E.M)) among all treatments, light–saturated cultures acclimated to 31°C were the only cultures that were able to maintained considerable $N_2$ fixation rates at assay temperatures as high as 34 °C (Fig. 3). In addition, both light availability and growth temperature affected the deactivation temperature ($T_h$) for $N_2$ fixation in *Trichodesmium* and no interaction between these two drivers was found on $T_h$ (Table 3). $T_h$ in light-saturated cultures tended to be higher than that in light-limited cultures regardless of the growth temperature. $T_h$ was lower in cultures grown at 31 °C compared to that in cultures grown at 23 or 27 °C under both light levels. The activation energy ($E_a$) for $N_2$ fixation was affected by growth temperature but not by growth light intensity (Table 3). The values of $E_a$ for $N_2$ fixation increased from 0.49 ± 0.04 eV (±S.E.M) to 0.91± 0.05 (±S.E.M) and 1.07± 0.04 eV(±S.E.M) as growth temperatures increased from 23 °C to 27 °C and 31 °C.

## 4 Discussion

In this study, light availability not only affected growth rate and $N_2$ fixation directly, but also modulated their responses to temperature changes in *Trichodesmium* IMS 101. Reduced energy supply due to light limitation leads to lowered nitrogen fixation and thus reduced growth in the diazotroph. The specific growth rate were maximal at 27 °C for saturating light–grown cells, but were virtually insensitive to temperature changes across the tested temperature (23, 27 and 31 °C) for light–limited cultures.

The interactions between temperature and light on *Trichodesmium* demonstrated in this work are relevant to natural light and temperature variations and to *Trichodesmium* global change physiology and biogeography. Light supplies energy for photosynthesis, growth and other key activities, such as $N_2$ fixation in cyanobacterial diazotrophs. The observed phenomenon that the growth rate becomes less sensitive to temperature changes (Fig 1(a) and Fig. 4) in *Trichodesmium* IMS 101 under limiting light levels can be attributed to insufficient energy supply for the cells to respond to temperature changes. While thermal biological responses are mainly based on enzymatic performance, light limitation suppresses syntheses of enzymes (Raven and Geider, 1988), and thus subsequently limits thermal responses. Although light–limited phytoplankton cells typically allocate more resources to light–harvesting systems to compensate for light shortages, at very low irradiances

this compensation cannot prevent light harvesting capacity from being a limiting factor for enzyme synthesis and growth

(Raven and Geider, 1988). Field investigations show that vertical distributions of *Trichodesmium* can reach to depths greater

than 100 m, where light is absolutely limiting and temperature is lower compared to surface temperature (Olson et al., 2015;

Rouco et al., 2016). According to the typical values of surface solar irradiances and vertical extinction coefficient in tropical

and subtropical oceans (Olson et al., 2015), the daily light dose received by the light–limited cultures in our study

corresponds to that received by *Trichodesmium* at a depth of 50–60 m. The contribution of biomass and $N_2$ fixation by

*Trichodesmium* at depths greater than 50 m might range from 7% to 28% (Davis and McGillicuddy, 2006; Olson et al.,

2015). Therefore, the evaluation of potential warming effects on *Trichodesmium* should not be constrained to the populations

inhabiting light–saturated environments (upper tens of meters) (Breitbarth et al., 2007; Jiang et al., 2018), making 3–

Dimensional models indispensable. In existing 3–Dimensional model studies involving *Trichodesmium* (Boyd and Doney,

2002; J. K. Moore et al., 2001), the combined effects of temperature and light on *Trichodesmium* biological activities are

simply assumed to be additive, which is proven to be inappropriate in this work. While the absolute values of $N_2$ fixation rate

under light limiting and saturating levels cannot be directly compared on the basis of Chl–a content, since lower light level

resulted in more cellular Chl–a content (Supplementary Fig. S1(b)), comparison of the thermal response patterns can

generate useful information for improving model predictions of diazotrophic responses to ocean climate changes.

Thermal responses for organisms are known to be useful in evaluating thermal acclimation potential and probing low and

high temperature tolerances (Gunderson et al, 2010; Somero, 2010; Way and Yamori, 2014). In this work, the shape of the

short–term thermal response curves for $N_2$ fixation is normalization–independent because cells were exposed to different

assay temperatures for only one hour, hardly changing the elemental stoichiometry or cellular pigments component. When

exposed to abrupt temperature gradients, the light-saturated cells acclimated to higher temperature exhibited higher $T_{opt}$

values (Table 3) and lower thermal susceptibility to supra–optimal temperatures ($E_h$; Table 3). This indicates an increased

capability for the diazotroph to tolerate short-term warming impacts. However, light limitation made the cells more

susceptible to warming due to decreased $T_{opt}$ and increased $E_h$ for $N_2$ fixation (Table 3). Moreover, with light limitation,

acclimation to high temperature did not help *Trichodesmium* cells tolerate short-term supral-optimal temperature.  On the

other hand, Chl–a fluorescence data shown that the PSII in light–limited cultures was as healthy as that in cells grown under

saturating light (Fig. 1(c), 2), and it has been shown that damage to PSII usually occurs at temperatures above 45 °C (Yamori et al., 2014). Therefore, the collapse of $N_2$ fixation at high temperature was not likely caused by the dysfunction of the photosystems, but might be caused by the uncoupling of adenosine triphosphate (ATP) synthesis to electron transport, since proton leakiness of the thylakoid membrane has been frequently proposed as a problem at high temperature (Yamori et al., 2014). This is consistent with the observation that supra–optimal temperature inhibition of $N_2$ fixation was aggravated by

light limitation (Fig. 3). In addition, damage to nitrogenase at high temperatures might also be one of the reasons responsible for the faster drop of $N_2$ fixation at high temperature in light–limited cultures (Gallon et a., 1993). This is because the extra investment of resources in repair of damaged nitrogenase could not be supported under light–limiting conditions (Fig. 3(b)). Therefore, light availability exerts critical control on the acclimation potential of $N_2$ fixation in *Trichodesmium* to warming.

Acclimation to different temperatures also affected the activation energy ($E_a$) for $N_2$ fixation in *Trichodesmium* IMS 101

(Table 3). For *Trichodesmium* species, $N_2$ fixation can be controlled by supply of ATP/reducing equivalents, mainly coming from photosynthesis, and the inherent catalytic capacity of the nitrogenase. These two processes may exhibit different temperature dependence, i.e. different $E_a$. The $E_a$ of the controlling process determines the $N_2$ fixation $E_a$ (Hikosaka, et al,, 2006; Staal et al., 2003). Therefore, the differences in $N_2$ fixation $E_a$ between cultures grown at different temperatures may reflect that $N_2$ fixation was primarily controlled by different processes in cultures acclimated to different temperatures.

Preliminary evidence supporting this hypothesis came from the various effects of assay light intensity on the values of $E_a$ for $N_2$ fixation between light–limited cultures grown at 23 °C and 27 °C (Supplementary Table S1, Fig. S2). For *Trichodesmium* grown under limiting light level, the lower $E_a$ values in cultures acclimated to 23 °C was significantly elevated by the increased assay light intensity which can provide more ATP/reducing equivalents (Supplementary Table S1; Fig. S2(a)). This suggests the constraint should be the supply of ATP/reducing equivalents. The higher $E_a$ values in cultures acclimated

to 27 °C were insensitive to the assay light intensity changes, suggesting $N_2$ fixation should not be controlled by the supply of ATP/reducing equivalents at this optimal temperature, but may possibly be controlled by inherent catalytic capacity of the nitrogenase (Supplementary Table S1; Fig. S2(b)).

The short–term thermal responses for $N_2$ fixation mirror thermal shock responses. If cells are exposed to the thermal changes for longer time, acclimation will definitely change the thermal responses for $N_2$ fixation in *Trichodesmium*

(Breitbarth et al., 2007; Fu et al., 2014; Staal et al., 2003). To compare the short–term and acclimated thermal responses for

$N_2$ fixation, we calculated the corresponding values of $E_a$, $E_h$ and $T_{opt}$, being respectively $0.93 \pm 0.64$ eV($\pm$S.E.M), $1.86 \pm$

$1.19$ eV($\pm$S.E.M) and $27.1 \pm 1.0$ °C($\pm$S.E.M), for fully–acclimated $N_2$ fixation within the range of 20–34 °C growth

temperatures in *Trichodesmium* IMS 101 (Breitbarth et al., 2007). These values of $E_a$ and $E_h$ are comparable to those derived

from short–term thermal response for $N_2$ fixation in the same strain grown under light–saturating condition and 31 °C in our

study (Table 3), but the $T_{opt}$ value is lower than that from short–term thermal response. On the other hand, we have tried to

derive values of $E_a$, $E_h$ and $T_{opt}$ for acclimated $N_2$ fixation rates in another three *Trichodesmium erythraeum* strains (strains

RLI, KO4-20 and 21-75) (Fu et al., 2014), but the model fitting failed to converge. Instead of been negatively skewed, the

thermal response curves of acclimated $N_2$ fixation in these three *Trichodesmium* strains are nearly symmetrical. These

comparisons show that thermal response for $N_2$ fixation in *Trichodesmium* are strains–specific, and are affected on the time

scale of acclimation process.

     In the oceans, *Trichodesmium* and other pelagic phytoplankton are often exposed to abrupt temperature changes due to

strongly disturbed weather conditions, such as tropical cyclones, and marine heat waves. Global warming has been predicted

to increase both tropical cyclone intensities, and the frequency of the most intense tropical cyclones (Elsner et al. 2008;

Knutson et al., 2010; Wehner et al., 2018). Upper ocean temperature declines prior and during cyclone event, and then

increases abruptly afterwards (Li et al., 2009), accompanied by strong variations of surface solar radiation and stratification

(Sriver and Huber, 2007). These abrupt temperature changes occurring in nature are not as acute as those in our experiment.

For example, temperature changes caused by cyclone and heat waves are on the scale of 0.5 - 1 °C per day (Babin et al.,

2004; Beca-Carretero et al., 2018). Nonetheless, these temperature changes occur within one generation of *Trichodesmium*

because of its low growth rate, leaving not enough time for full acclimation. Therefore, the values of $E_a$, $E_h$, $T_h$ and $T_{opt}$

provided in this study can likely serve as proxies for some types of abrupt natural temperature increases.

**Code/Data availability**

All data obtained in this study are in Supplement.

**Author contribution**

KG and XY designed the experiment. XY carried out the experiment. XY, FXF, DH and KG analysed the data and wrote the
manuscript.

**Competing interests**

The authors declare no competing of interest.

**Acknowledgements**

This study was supported by the National Key R & D Program of China (2016YFA0601400), National Natural Science
Foundation of China (41720104005, 41721005), and by U.S. National Science Foundation grants OCE 1538525, OCE
1657757, OCE 1638804 and OCE 1851222. The authors declare no conflict of interest.

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

**Table 1** Results of two-way ANOVA for specific growth rate, $N_2$ fixation rate, effective quantum yield and parameters derived from rETR light curve with interactions between "Temperature" and "Light". See Supplementary Table S2 for results of pair-wise comparisons.

| Parameter | Effect | d.f | F value | P value |
|---|---|---|---|---|
| Specific growth rate | **Temperature** | **2,12** | **22.0** | **< 0.001** |
| | **Light** | **1,12** | **662.7** | **< 0.001** |
| | **Temperature*Light** | **2,12** | **18.0** | **< 0.001** |
| $N_2$ fixation rate ($N_{growth}$) | **Temperature** | **2,12** | **3912.3** | **< 0.001** |
| | **Light** | **1,12** | **149.9** | **< 0.001** |
| | **Temperature*Light** | **2,12** | **112.7** | **< 0.001** |
| Effective quantum yield | **Temperature** | **2,12** | **22.1** | **< 0.001** |
| | **Light** | **1,12** | **233.2** | **< 0.001** |
| | Temperature*Light | 2,12 | 1.8 | 0.211 |
| alpha | **Temperature** | **2,12** | **24.5** | **< 0.001** |
| | **Light** | **1,12** | **10.6** | **< 0.01** |
| | Temperature*Light | 2,12 | 0.2 | 0.815 |
| $E_k$ | **Temperature** | **2,12** | **5.0** | **< 0.05** |
| rETR light curve | **Light** | **1,12** | **6.0** | **< 0.05** |
| | Temperature*Light | 2,12 | 1.0 | 0.394 |
| $rETR_{max}$ | **Temperature** | **2,12** | **31.2** | **< 0.001** |
| | **Light** | **1,12** | **139.8** | **< 0.001** |
| | **Temperature*Light** | **2,12** | **8.1** | **< 0.01** |


**Table 2** The light harvesting efficiency ($\alpha$), relative election transport rate maximum (rETR$_{max}$) and light saturation point ($E_k$), derived from the light curves (Fig. 2), for *Trichodesmium* grown at different temperature and light intensity levels; values represent the means ± standard deviations of biological replicates (n=3); superscripts with different letters represent significant difference (Turkey's test, more details in Supplementary Table S2; $p<0.05$;) among the treatments. The units of $E_k$ and rETR$_{max}$ are μmol quanta m$^{-2}$ s$^{-1}$ and arbitrary unit, respectively.

| | Growth conditions | | | | | |
|---|---|---|---|---|---|---|
| | Light-saturating | | | Light-limiting | | |
| | 23 °C | 27 °C | 31 °C | 23 °C | 27 °C | 31 °C |
| $\alpha$ | $0.25 \pm 0.01^{ac}$ | $0.24 \pm 0.03^{a}$ | $0.28 \pm 0.01^{c}$ | $0.30 \pm 0.03^{bc}$ | $0.28 \pm 0.03^{b}$ | $0.35 \pm 0.03^{b}$ |
| $E_k$ | $316 \pm 22^{ab}$ | $322 \pm 45^{ab}$ | $371 \pm 16^{a}$ | $270\pm 17^{b}$ | $319 \pm 38^{ab}$ | $329 \pm 21^{ab}$ |
| rETR$_{max}$ | $78 \pm 3^{a}$ | $72 \pm 3^{a}$ | $105 \pm 2^{b}$ | $80 \pm 6^{a}$ | $90 \pm 2^{c}$ | $115 \pm 4^{b}$ |

**Table 3** Model parameters of thermal responses for $N_2$ fixation. The structure of the fixed effect is: $N(T_c)$ ~ Temperature * Light; $E_a$ ~ Temperature; $E_h$ ~ Light + Temperature; $T_h$ ~ Light + Temperature. "+" and "*" represent additive and interactive effects, respectively.

| Parameter | Light | Temperature(°C) | Estimate | S.E.M | CI(95%) |
|---|---|---|---|---|---|
| $N(T_c)$ ($\mu$mol $N_2$ mg Chl-a$^{-1}$ h$^{-1}$) | Light-saturating | 23 | 38.3 | 1.0 | [36.4, 40.3] |
| | | 27 | 39.1 | 1.0 | [37.1, 41.1] |
| | | 31 | 41.3 | 1.5 | [38.2, 44.3] |
| | Light-limiting | 23 | 19.5 | 0.8 | [18.0, 21.1] |
| | | 27 | 15.0 | 0.8 | [13.4, 16.6] |
| | | 31 | 20.3 | 1.1 | [18.2, 22.5] |
| $E_a$ (eV) | No Light effect | 23 | 0.49 | 0.04 | [0.41, 0.57] |
| | | 27 | 0.91 | 0.05 | [0.80, 1.01] |
| | | 31 | 1.07 | 0.04 | [0.98, 1.16] |
| $E_h$ (eV) | Light-saturating | 23 | 4.49 | 0.51 | [3.47, 5.51] |
| | | 27 | 3.99 | 0.31 | [3.36, 4.61] |
| | | 31 | 1.47 | 0.14 | [1.18, 1.75] |
| | Light-limiting | 23 | 7.51 | 0.68 | [6.15, 8.87] |
| | | 27 | 7.01 | 0.60 | [5.82, 8.21] |
| | | 31 | 4.49 | 0.49 | [3.50, 5.48] |
| $T_h$ (°C) | Light-saturating | 23 | 32.6 | 0.1 | [32.3, 32.9] |
| | | 27 | 32.4 | 0.2 | [32.1, 32.8] |
| | | 31 | 31.8 | 0.2 | [31.3, 32.3] |
| | Light-limiting | 23 | 31.1 | 0.1 | [30.9, 31.4] |
| | | 27 | 31.0 | 0.1 | [30.7, 31.2] |
| | | 31 | 30.3 | 0.2 | [29.9, 30.6] |
| $T_{opt}$ (°C) | Light-saturating | 23 | 28.7 | 0.2 | [28.2, 29.1] |
| | | 27 | 29.5 | 0.2 | [29.2, 29.8] |
| | | 31 | 30.0 | 0.3 | [29.5, 30.6] |
| | Light-limiting | 23 | 28.2 | 0.2 | [27.9, 28.6] |
| | | 27 | 28.6 | 0.2 | [28.3, 28.9] |
| | | 31 | 27.8 | 0.2 | [27.4, 28.2] |
| $N_{max}$ ($\mu$mol $N_2$ mg Chl-a$^{-1}$ h$^{-1}$) | Light-saturating | 23 | 16.0 | 0.3 | [15.3, 16.6] |
| | | 27 | 19.3 | 0.4 | [18.5, 20.1] |
| | | 31 | 14.6 | 0.4 | [13.9, 15.4] |
| | Light-limiting | 23 | 8.3 | 0.3 | [7.6, 8.9] |
| | | 27 | 7.4 | 0.4 | [6.6, 8.2] |
| | | 31 | 8.6 | 0.4 | [7.8, 9.4] |



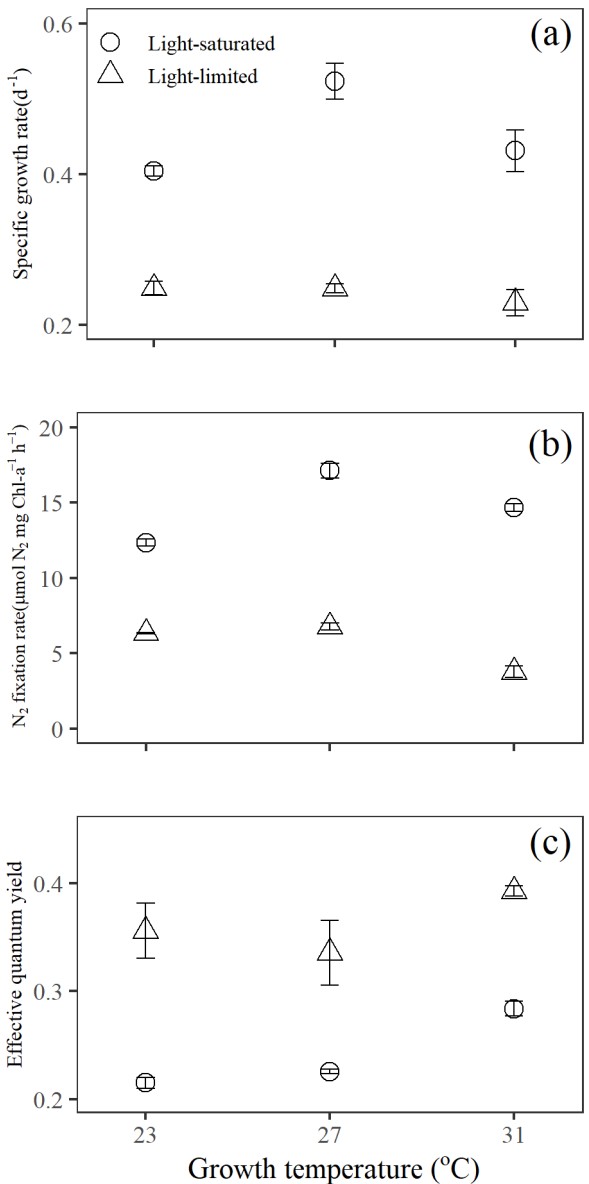

**Figure 1** *Trichodesmium* responses of (a) growth, (b) N$_2$ fixation rate and (c) effective quantum yield to temperature and light availability interactions; values represent the means ± the standard deviations of biological replicates(n=3).

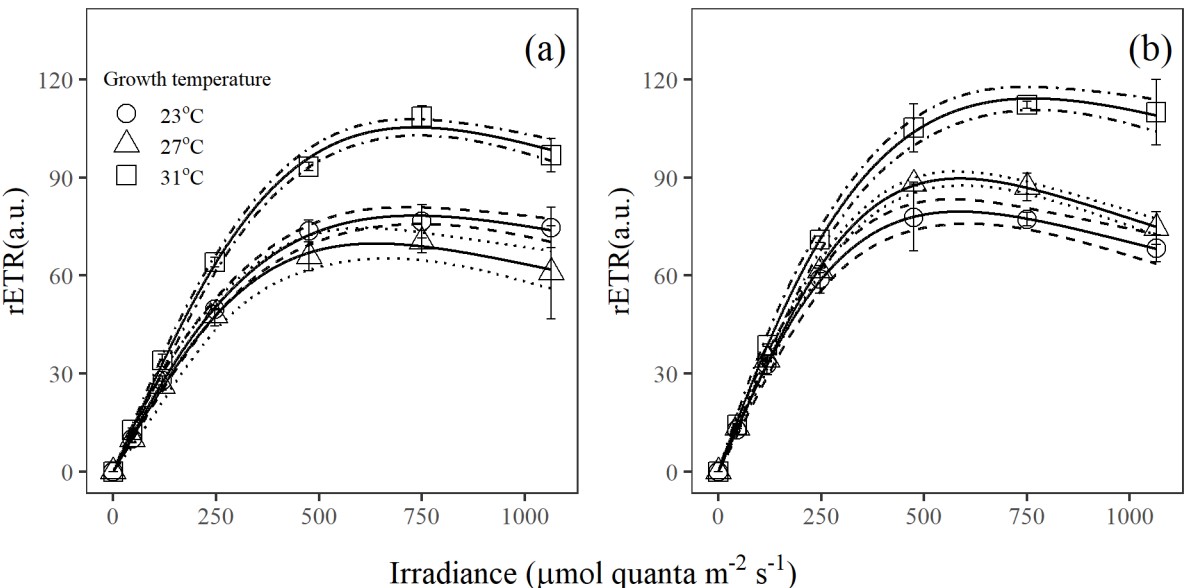

**Figure 2** Light response curves of rETR in *Trichodesmium* populations grown under (a) light–saturating and (b) light–limiting conditions; values represent the means ± standard deviations of biological replicates(n=3); Solid lines illustrate the best fit to Eq. (4) with 95% confidence intervals as dashed lines.

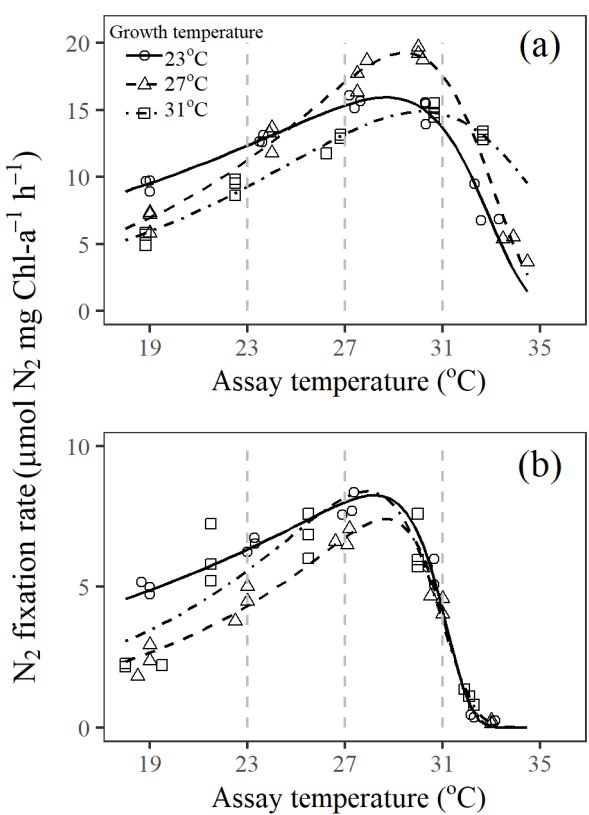


**Figure 3** Short–term thermal response curves for N$_2$ fixation rate in *Trichodesmium* cultures grown under (a) light–saturating and (b) light–limiting conditions; fitted lines are based on fixed effect coefficients of the nonlinear mixed effects model fitting to Eq. (2); vertical dotted lines mark the assay temperatures 23 °C, 27 °C and 31 °C.


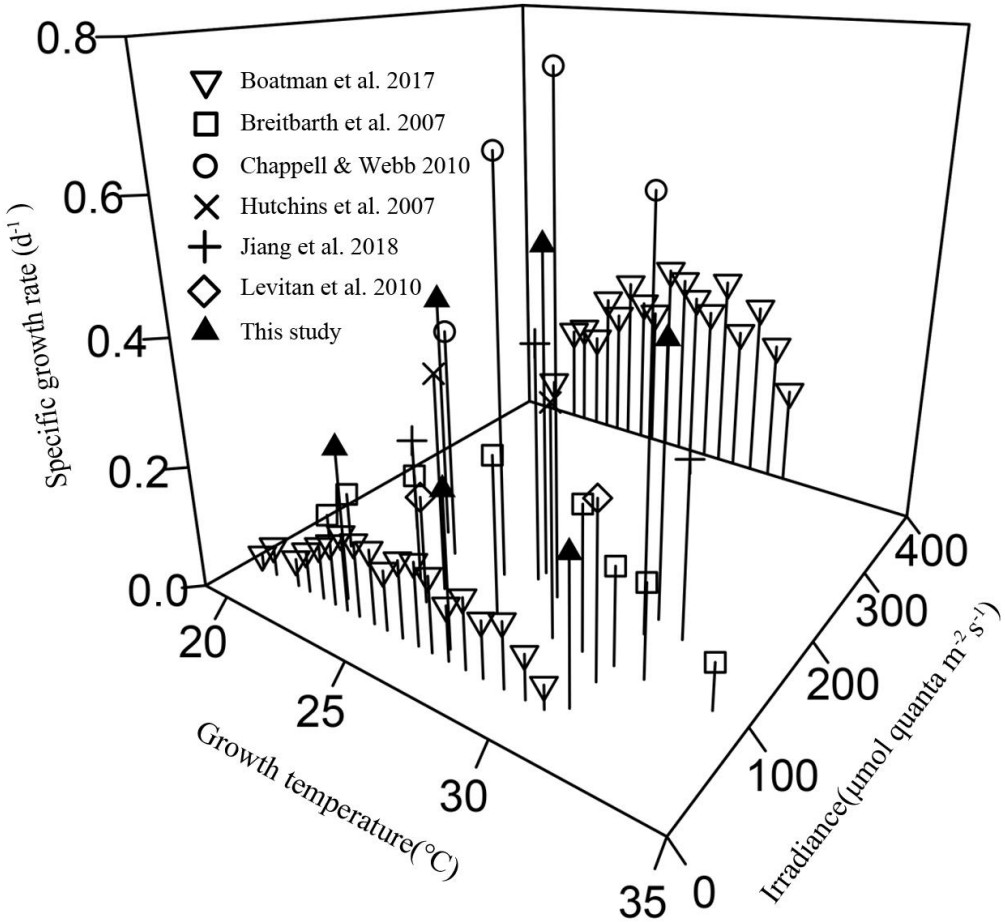

**Figure 4** The combined effects of temperature and light intensity on the specific growth rate in *Trichodesmium* IMS 101; data

from published literatures involving at least two growth temperatures and this study.