# Peer review of "Light availability modulates the effects of warming in a marine $N_2$ fixer"

_Biogeosciences, 2019_

## Referee Comment (RC1) · Anonymous Referee #1 · 5 Dec 2019

The authors investigated the combined effects of light and temperature on the growth, N2 fixation and photosynthesis in the marine diazotroph, Trichodesmium. Light and temperature are two of the most environmental drivers for this species as for other marine primary producers. However, the combined effects of these two factors have surprisingly little been documented on Trichodesmium. This work fills such gap. The new finding from this work is that the thermal responses in Trichodesmium are strongly dependent on light exposures when grown under different light and temp levels. The parameters derived from the measurement are of significance in predicting the responses of Trichodesmium to ocean physical environmental changes associated with global changes. Generally, this work has been well performed and delivers a clear message, but some revisions are needed before being considered acceptable for pub-

lication at BG:

1. Line 65, ". . . where light intensity could be as low as 2 $\mu$mol quanta m-2 s-1". What's the source of this number? 2. Line 69, ". . . Trichodesmium's N2 fixation and growth,". It's better to delete " 's N2 fixation and growth". 3. Line 115 – 118. In the treatment "light-limiting, 31 oC", the N2 fixation rate under growth condition was obtained through an indirect and unusual way. I recommend that the authors should also take the N2 fixation rate measured at >31oC into consideration (maybe use the average of this and that measured at 30 oC), although such modification may alter the Figure 1b, and require revision of related text. 4. Line 122. ". . .Aliquots of 1.5 m . . ." should be "1.5 ml". 5. The authors should describe the statistical analysis techniques they used in the Material and methods. Although I can roughly deduce the used statistical techniques from the text in Results, the authors should explicitly present them, which will help readers evaluate their results and conclusions. 6. Figure 3. It seems that the selections of temperature gradients are different among different treatments, which is uncommon. Why? Will this affect the interpretation of the data? 7. Line 202-205. How did the authors get the numbers ">28% and 7%-20%"? The cited literatures do not provide such numbers. 8. Table 1. In the text, the light treatments were referred as "light limiting" and "light saturating", but in this Table they were denoted as "LL" and "HL". It will be better to keep them consistent. 9. Fig 3b. The temperature norm of N2 fixation in the treatment "light-limiting, 31 oC" is quite different from those in other treatments, which deserves more discussion. However, authors didn't put much attention on this phenomenon.

All in all, this work focused on a valuable but previously overlooked scientific topic and obtained some interesting results. If the authors can properly deal with the concerns listed above, I think it will be qualified to be published in BG.

---

## Referee Comment (RC2) · Anonymous Referee #2 · 5 Dec 2019

General comments: This manuscript by Yi et al. examines how light availability (tested at two levels of light intensity) interacts with the effects of warming (along a gradient of three temperatures) in a marine N2 fixer (Trichodesmium erythraeum IMS101) across a time scale of about ten generations. The experiment is in its essence a two-driver question, where either driver might intrinsically decrease or increase metabolic performance, but the cumulative effect is unknown. The findings and the results are straightforward, with a clearly identifiable general trend. While theoretically relevant (e.g. changes in temperature may coincide with changes in light intensity), it is not quite clear why the authors chose these two drivers over other sets of drivers until much later on in the manuscript. It would also have been nice to see a more explicit evaluation over whether the changes in temperature/light level constitutes an environ-

mental deterioration or amelioration and how that impacts on how they interact. Still, the results are quite interesting, especially since they cover a range of phenotypic traits (growth rates, N2 fixation rates, photosynthetic machinery). However, I have major concerns about how the results are presented: the methods do not indicate how the data were analysed, and the results appear largely as post-hoc output. The latter would indicate that the authors used an ANOVA or similar test, which is indeed indicated more clearly once in line 185, but details are nowhere to be found. For example, a statement about the data is followed up simply by (p<0.05,tukey HSD method). It is impossible to glean from this what kind of data were compared and what the original model looked like. As the main question is about interactive effects, and the data are hierarchical in nature (e.g. differently acclimated samples used in a short-term assay), the authors would have needed some kind of mixed model approach. The closest the text ever gets to describing how the data were handled is in line 129 'parameters can be obtained through non-linear least squares regression in R language'. Which packages did the authors use to do so? How did they fit their data to the Eiler curve? Similarly, the authors mention the Sharpe-Schoolfield model, but that would be no easy feat with only 5 temperatures (it is a 4 parameter equation). More information would have been crucial here! It clearly worked well, as the fits in Figure 3 don't look too bad. However, we then need to also know how different these curves are from each other. For this, one needs to either extract the parameters and compare them (and describe how!) or run a non-linear mixed effects model (and describe how). As it stands, the handling and analysis of data is not at all traceable. I will provide suggestions on how to deal with this issue in the detailed comments below.

Technical comments and corrections, further suggestions:

Throughout: please double-check use of singular/plural and use of present tense and past tense. Please be careful with the vocabulary used. What is 'acclimation', what is 'short term'? How are either of these different from 'acute'? Be consistent throughout in how you use these words. You could, for example, define them in the introduction

and then stick to that definition.

Abstract Line 13: Consider telling the reader which phenotypes from the get go. Line 16: 'range of 23-31' could be misleading, just state the three temperatures Line 16/17: 'when the acclimation . . . [. . .]. . . to growth temperature was evaluated by short-term

Line 22: "cells growing under low light levels while distributed deep in the euphotic zone or under cloudy weather conditions might be more susceptible to ocean warming": I would be careful about that, the study refers to response of acclimated cells at different conditions, not to acute or immediate responses (at least for the growth response), especially when we consider that these cells can actively migrate along the water column.

Line 23: Point out explicitly that this is true for ocean warming occurring on the timescales of a few generations, or, as in your assays, short term responses within the same generation in mere hours. Mention scenarios when this is applicable upfront (mixing, heat waves..)

Introduction Line 29: might not be all that 'obvious' to all readers. Consider elaborating. Line 39: The 1960s are not a century ago yet, plus the literature cited after this statement is pretty recent? Specifically: Is there a reference for the 1960 discovery of diazotrophy in Trichodesmium?

Line 41/42: 'In the IPCC. . .[. . .]' consider rephrasing to 'The IPCC scenario [. . .] predicts..[..]' Line 43: I am not sure Collins et al 2013 is the correct reference here, as it is focused on the long term implications of global climate change, not so much the ocean physics Line 44: 'consequences' on what? Consider elaborating.

Line 50-52: different responses to warming more due to relation between traits and environment, than only "because of the spatial heterogeneity of present temperatures and projected warming". Clarify it is also a matter of local adaptation.

Line 68: clearly state that Trichodesmium is ACTIVELY able to migrate vertically.

Methods:

Line 75: Are three replicate populations enough to assess within species variability? Was this decision based on pilot studies? Were the cultures clonal? Mixed?

Line 77: Would be crucial to know where these three temperatures lie on the thermal tolerance/performance curve. The 2007 and 2014 studies just state that these are temperatures that this specific Trichodesmium can live in?

Line 77: Might have been better to have used a third light intensity toward the Iopt, just for the sake of comparison and to underpin the basic response to temperature of Trichodesmium.

Line 77: 160 $\mu$mol quanta m-2 s-1 seem like quite a low light intensity to be saturating, although they report in the supplementary a pilot study that seems confirm the statement. Nevertheless, the cultures for the pilot study were not aerated while it seems to be a constant for Trichodesmium culturing in all other papers (formation of cells′ aggregates and consequently maybe self-shading effects?).

Line 84: 'cyanobacteria were floating singly' consider rephrasing to 'cyanobacteria floated as single filaments' Line 85: Was there a round of pre-acclimation prior to the acclimation phase? Pre-acclimation is a crucial step to avoid carry-over from the previous culture conditions. See for example Trimborn et al 2019, Front. Mar. Sci https://doi.org/10.3389/fmars.2019.00167, Schaum and Collins 2014, Proc Biol Sci.281(1793): 20141486, Scheinin et al 2015 https://doi.org/10.1098/rsif.2015.0056, Lenski 2017 The ISME Journal volume 11, pages2181–2194(2017)

Line 86: How were the growth rate curves fitted? Missing info

Line 94: should be 'before applying the natural logarithm' instead of ' before natural logarithm'. Generally, how does using Chla as a proxy for growth deal with cells having more Chla per cell? Line 99: 'acute' as stated above, be mindful of vocabulary used. Define once, then stick to it. Line 102: is 0.5 to read 50 minutes or 30 minutes? This

seems really short for a 25mL vial to equilibrate to the correct temperatures! Line 107: The Padfield paper is pivotal, but it is not about the Schoolfield equation per se (it is about adaptation to warming and uses the Schoolfield as a tool).The second correct reference is Sharpe, P. J. & DeMichele, D. W. Reaction kinetics of poikilotherm development. J. Theor. Biol. 64, 649–670 (1977).

Line 113: Which package was used for the "optimize" function? Which version?

Line 114: If used correctly, the Sharpe-Schoolfield output should not require the 'optimize' function, but simply, rates at Topt can be obtained by re-arranging the equation. It is really not clear at all here how the data were fitted to the Sharpe-Schoolfield (it clearly went well as the figure looks correct). To me, it would make sense to either extract the parameters (Ea, Eh, Topt. Tc) and then compare them via a mixed model (e.g. parameter $\sim$ growthtemp*light with replicate within treatment as the random effect) or fit a non-linear mixed effects model where lnNrate $\sim$ schoolfield.high(ln.c,Ea,Eh,Th,temp=K,Tc=your Tc value) and, to begin with fixed = list(ln.c + Ea + Eh + Th $\sim$ growthtemp*light). You can then compare AICcs of your models (e.g. test also additive effect, each on their own, and just the intercept) and chose the best one. If you compare extracted parameter values, then the MuMin dredge function will come in handy!

Line 116: Why was it not possible to measure N2for samples at 31° C? At what time were the samples taken? I know N2 fixation-related genes show a strong circadian cycle, maybe a similar mechanism is involved?

As the authors stated into the nice small "meta-analysis", there is a huge within strains variation, why don't you used more strains? Alternatively, more isolates instead of three if you wanted to assess for within strains variations?

Line 129: See comments above – how were the data dealt with? Again, you can either extract parameters and compare via a mixed model, or run a non-linear mixed model starting with the most complex model and then working your way down to the

most simple model. For all other phenotypic traits (the ones where you are not fitting a slope) , a mixed model seams the way to go!

Results: Throughout: When giving a value, also give the standard deviation or standard error. When referring to the result of statistical test, just giving the post-hoc value is not enough, as that only refers to ONE specific pair-wise comparison. If reporting one specific pair-wise comparison, we need to know which one!

Line 140: Might be worth starting out with whether the combined effect of light limitation was indeed interactive, or additive, or if one out of the two described the data best. Without the appropriate reporting of the stats involved, this is impossible to tell. Line 141: see above. Strictly speaking, this is not a temperature range, but three temperatures, 23,27,and 31oC. Line 145: How much is 'slightly'? Line 152: How much higher is higher? Line 159: Is acute the same as short-term here? Pick a word, then stick to it. Line 168: What was the variation around this 1.4 oC increase? Line 178: be mindful of the tense. Should be 'were able to sustain' Line 183: Add SD or standard error to these values Line 185: again, not clear what the p value refers to, or what was actually tested in the two way ANOVA

Discussion Line 191: "negative growth effects" seems a strong statement, maybe better use "reduced"

Line 196: level should be levels

Line 202: "temperature is lower" than surface?

Line 206: maybe I didn't get it, but "respectively" to what?

Line 210: This is a very nice and clear summary (the additive vs interactive bit), however, without the correct statistical approach it is impossible to tell whether the data actually support this conclusion! Line 232: May need a reference here Line 235: Should be equivalents, not equivalent

Line 250: what is the difference here between 'acclimated" and "short-term"? You

mentioned both "short-term temperature norms" and "acclimation" throughout the paper (e.g. Table S1). Please clarify.

Line 257: 'a bit different' is too vague Line 258: not sure if 'and/or' is the correct choice of words here. Plus, it should be 'on the time scales of acclimation processes' . Consider adding that here, this is approximately 10 generations. Line 259: What about within-strain variation? Line 266: 'to some extent' is a bit vague, may need a bit more information here.

Tables Spell out HL and LL as high light and low light You clearly have the data from the light curves in the table, so explaining how you actually got them should not cause too much agony (we hope).

Figures Might be worth mentioning the software the figures were produced in.

Figure 1 The lettering of the subpanels as a, b,c, is highly confusing with the significance levels using the same lettering. Might be easier to present the significance levels as a table? What are the slopes in this graph? How were they fitted?

Figure 2 Spell out what a.u. stands for. Consider adding confidence intervals to model fits

Figure 4 Not clear where the interactions are. Again, the significance levels are a bit distracting and probably better displayed in a table.

Figure 5: a) Probably good idea to highlight the symbol for this study in bold b) –d) why are there no SDs or confidence intervals?

[Figure]

---

## Author Comment (AC1) · 24 Dec 2019

The authors investigated the combined effects of light and temperature on the growth, N2 fixation and photosynthesis in the marine diazotroph, Trichodesmium. Light and temperature are two of the most environmental drivers for this species as for other marine primary producers. However, the combined effects of these two factors have surprisingly little been documented on Trichodesmium. This work fills such gap. The new finding from this work is that the thermal responses in Trichodesmium are strongly dependent on light exposures when grown under different light and temp levels. The parameters derived from the measurement are of significance in predicting the responses of Trichodesmium to ocean physical environmental changes associated with global changes. Generally, this work has been well performed and delivers a clear message, but some revisions are needed before being considered acceptable for publication at BG:

1. Line 65, ". . . where light intensity could be as low as 2 μmol quanta m-2 s-1". What's the source of this number?

**Response: We used the following equation to get this number:**

$$E(d) = E0 * exp(-k * d)$$

**$E(d)$ is the light intensity (μmol quanta $m^{-2}$ $s^{-1}$) at depth $d(m)$; $k$ is the light extinction coefficient; $E0$ is the surface solar irradiance. We assumed that the water column was homogenous, extinction coefficient was 0.05 $m^{-1}$ (common value reported for subtropical and tropical pelagic oceans (Olson et al., 2015)) and surface solar irradiance was 2000 μmol quanta $m^{-2}$ $s^{-1}$.**

**In the revised manuscript, we will add references here which can directly give similar number.**

2. Line 69, ". . . Trichodesmium's N2 fixation and growth,". It's better to delete " 's N2 fixation and growth".

**Response: we will follow this suggestion.**

3. Line 115 – 118. In the treatment "light-limiting, 31 oC", the N2 fixation rate under growth condition was obtained through an indirect and unusual way. I recommend that the authors should also take the N2 fixation rate measured at >31oC into consideration (maybe use the average of this        and that measured at 30 oC), although such modification may   alter the Figure 1b,        and require revision of related text.

**Response: We will try this.**

4.   Line 122.   ". . .Aliquots of 1.5 m . . ." should       be "1.5 ml".

**Response: This will be corrected.**

5.   The authors should describe the statistical analysis techniques they used in the Material and methods. Although I can roughly deduce the used statistical techniques from the text in Results, the authors should explicitly present them, which will help readers evaluate their results and conclusions.

**Response: This was a big mistake, and we will add the paragraphs describing how we analyze the data. Generally, we used the two-way ANOVA and Tukey test to determine the effects of light, temperature and interaction of light and temperature on the tested physiological traits, such as growth rate, effective photochemical efficiency, $N_2$ fixation rate (Figure 1) and parameters obtained through non-linear fitting (Table 1; Figure 2, 3, 4).**

6. Figure 3. It seems that the selections of temperature gradients are different among different treatments, which is uncommon. Why? Will this affect the interpretation of the data?

**Response: We found that the temperature was not homogeneous in the multi-zone chambers that were used to measure the response of $N_2$ fixation to acute temperature changes (Figure 3), so we used the actually measured temperatures rather than the pre-set temperatures. No, this should not be a problem.**

7. Line 202-205. How   did the authors get the numbers ">28% and 7%-20%"? The cited literatures do not provide such numbers.

**Response: We got these numbers from the figures in the cited references (Figure 3 in Davis & McGillicuddy, 2006; Fig 8 and 10 in Olson et al., 2015), although they do not show up in the text.**

8.   Table 1.   In the text, the light treatments were referred as   "light limiting" and "light saturating", but in this Table they were denoted as "LL" and "HL".   It will be better to keep them consistent.

**Response: We will replace "LL" and "HL" with "light limiting" and "light saturating".**

9. Fig 3b. The temperature norm of N2 fixation in the treatment "light-limiting, 31 oC" is quite different from those in other treatments, which deserves more discussion. However, authors didn't put much attention on this phenomenon.

**Response: We guess that the unusual performance in treatment "light-limiting, 31 °C" might be related to the nitrogenase damage which was induced by the high growth temperature and exacerbated by the light limitation. The quantity of the functional nitrogenase might be not enough to form the expected $N_2$ fixation peak. We will try to explain this phenomenon in the revised manuscript.**

All in all, this work focused on a valuable but previously overlooked scientific topic and obtained some interesting results. If the authors can properly deal with the concerns listed above, I think it will be qualified to be published in BG.

Davis, C. S., & McGillicuddy, D. J. (2006). Transatlantic Abundance of the $N_2$-Fixing Colonial Cyanobacterium *Trichodesmium*. *Science,* 312(5779), 1517-1520. doi:10.1126/science.1123570

Olson, E. M., McGillicuddy, D. J., Flierl, G. R., et al. (2015). Mesoscale Eddies and Trichodesmium Spp. Distributions in the Southwestern North Atlantic. Journal of Geophysical Research: Oceans, 120(6), 4129-4150. doi:10.1002/2015JC010728

---

## Author Comment (AC2) · 24 Dec 2019

General comments: This manuscript by Yi et al. examines how light availability (tested at two levels of light intensity) interacts with the effects of warming (along a gradient of three temperatures) in a marine N2 fixer (Trichodesmium erythraeum IMS101) across a time scale of about ten generations. The experiment is in its essence a two- driver question, where either driver might intrinsically decrease or increase metabolic performance, but the cumulative effect is unknown. The findings and the results are straightforward, with a clearly identifiable general trend. While theoretically relevant (e.g. changes in temperature may coincide with changes in light intensity), it is not quite clear why the authors chose these two drivers over other sets of drivers until much later on in the manuscript. It would also have been nice to see a more explicit evaluation over whether the changes in temperature/light level constitutes an environmental deterioration or amelioration and how that impacts on how they interact. Still, the results are quite interesting, especially since they cover a range of phenotypic traits (growth rates, N2 fixation rates, photosynthetic machinery). However, I have major concerns about how the results are presented: the methods do not indicate how the data were analysed, and the results appear largely as post-hoc output. The latter would indicate that the authors used an ANOVA or similar test, which is indeed indicated more clearly once in line 185, but details are nowhere to be found. For example, a statement about the data is followed up simply by (p<0.05, tukey HSD method). It is impossible to glean

from this what kind of data were compared and what the original model looked like. As the main question is about interactive effects, and the data are hierarchical in nature (e.g. differently acclimated samples used in a short-term assay), the authors would have needed some kind of mixed model approach. The closest the text ever gets to describing how the data were handled is in line 129 'parameters can be obtained through non-linear least squares regression in R language'. Which packages did the authors use to do so? How did they fit their data to the Eiler curve? Similarly, the authors mention the Sharpe-Schoolfield model, but that would be no easy feat with only 5 temperatures (it is a 4 parameter equation). More information would have been crucial here! It clearly worked well, as the fits in Figure 3 don't look too bad. However, we then need to also know how different these curves are from each other. For this, one needs to either extract the parameters and compare them (and describe how!) or run a non-linear mixed effects model (and describe how). As it stands, the handling and analysis of data is not at all traceable. I will provide suggestions on how to deal with this issue in the detailed comments below.

**Response:**

**We are grateful for the referee's constructive comments and suggestions on our manuscript. We have studied them carefully.**

**As the referee points out, it would be better if we had explained why we chose light and temperature over other drivers at the beginning of the manuscript. We will revise the Introduction to handle this issue.**

**It was a serious mistake that we have omitted the crucial paragraphs describing how we analyzed the data. We performed the two-way ANOVA with normality (Shapiro-Wilk test) and variance homogeneity (Levene's test) tests to determine whether light, temperature and the interaction of light and temperature affected the phenotypic traits (Figure 1), including growth rate, effective photochemical efficiency and $N_2$ fixation rate. Then, post hoc (Tukey) test was used to do the pairwise comparisons. As with the data in Table 1 and Figure 2, 3, 4, we first extracted the parameters from the non-linear fitting to individual measurement. Then, the two-way ANOVA and Turkey test were used to determine the effects of light, temperature and their interaction on these parameters. The**

data analysis was done using the R language (version 3.5.3) with the built-in functions, including 'aov', 'shapiro.test' and 'TukeyHSD', function 'nlsLM' from package 'minpack.lm (version 1.2-1)' (line 113) and function 'leveneTest' from package 'car (version 3.0-2)'. We argue that our data analysis processes were appropriate for most of the tested physiological traits. Also, these statistical methods are widely used in other similar work, such as (Hong et al., 2017; Hoppe et al., 2018; Trimborn et al., 2019). Hoverer, we agree with the referee that the part involving the Sharpe-Schoolfield model (Figure 3, 4) might be problematic. Using 5 data points to fit a 4-parameter equation is overparameterized. We are grateful that the referee suggests an alternative statistical method to handle this problem, that is, non-linear mixed effects model. We have tried to use this method to re-analyze our data. The preliminary results are promising and do not change our main results and conclusions.

In the revised manuscript, we will add the paragraphs describing how we analyze our data and present the results in a more traceable way.

Technical comments and corrections, further suggestions:

Throughout: please double-check use of singular/plural and use of present tense and past tense.

Response: We will double-check and correct the syntax errors.

Please be careful with the vocabulary used. What is 'acclimation', what is 'short term'? How are either of these different from 'acute'? Be consistent throughout in how you use these words. You could, for example, define them in the introduction and then stick to that definition.

Response: In our manuscript, "acclimation" means that the cells are kept in a certain growth condition for more than 10 generations and their growth rates are stable. "acute" and "short-term" are interchangeable, referring to processes that occur within several hours.

We will define these terms in the revised manuscript.

Abstract Line 13: Consider telling the reader which phenotypes from the get go. Line 16: 'range of 23-31' could be misleading, just state the three temperatures Line 16/17: 'when the acclimation … […]… to growth temperature was evaluated by short-term

**Response: We will revise the manuscript accordingly.**

Line 22: "cells growing under low light levels while distributed deep in the euphotic zone or under cloudy weather conditions might be more susceptible to ocean warming": I would be careful about that, the study refers to response of acclimated cells at different conditions, not to acute or immediate responses (at least for the growth response), especially when we consider that these cells can actively migrate along the water column.

Line 23: Point out explicitly that this is true for ocean warming occurring on the timescales of a few generations, or, as in your assays, short term responses within the same generation in mere hours. Mention scenarios when this is applicable upfront (mixing, heat waves..)

**Response: We measured such phenotypical traits as growth, $N_2$ fixation, effective photochemical efficiency of *Trichodesmium* cultures acclimated to different light intensity and temperature levels. Additionally, we also measured the response of $N_2$ fixation to acute temperature changes. Our data directly indicate that light limitation leads to lower sensitivity of acclimated growth rate and N fixation to temperature change (Figure 1). On the other hand, the light-limited *Trichodesmium* cultures might be more vulnerable to acute warming (on the time scale of hours) in terms of $N_2$ fixation (Figure 3, 4). These two parts are related to different but interrelated scenarios. The former is related to the long-term environment changes, such as global warming, and the latter is more related to strongly disturbed weather conditions, such as cyclones, and heat waves. Studies show that the strong cyclones will be more frequent and stronger in the warmer oceans (Elsner et al., 2008; Knutson et al., 2010; Wehner et al., 2018).**

**We will revise these sentences to clarify the ambiguity.**

Introduction Line 29: might not be all that 'obvious' to all readers. Consider elaborating.

**Response: We will follow this suggestion to make the "obvious" really obvious.**

Line 39: The 1960s are not a century ago yet, plus the literature cited after this statement is pretty recent? Specifically: Is there a reference for the 1960 discovery of diazotrophy in Trichodesmium?

**Response: Modern interest in *Trichodesmium* dates to the 1960s with the recognition that *Trichodesmium* is diazotrophic.**

**Yes, (Dugdale et al., 1964; Dugdale et al., 1961; Goering et al., 1966). We will revise this part and cite the original papers.**

Line 41/42: 'In the IPCC...[...]' consider rephrasing to 'The IPCC scenario [...] predicts..[..]'

**Response: We will do this.**

Line 43: I am not sure Collins et al 2013 is the correct reference here, as it is focused on the long term implications of global climate change, not so much the ocean physics

**Response: The acclimated phenotypic traits, such as growth rate, $N_2$ fixation rate etc., are related to this reference. Superposed on this, we also measured the response of $N_2$ fixation to acute temperature change, which is more related to strong disturbed weather conditions.**

Line 44: 'consequences' on what? Consider elaborating.

**Response: Such as the primary production, carbon sequestration, elements cycles etc. We will elaborate this in the revised manuscript.**

Line 50-52: different responses to warming more due to relation between traits and

environment, than only "because of the spatial heterogeneity of present temperatures and projected warming". Clarify it is also a matter of local adaptation.

**Response: Yes. Local adaptation is another factor affecting organism's response to climate change. We will add this.**

Line 68: clearly state that Trichodesmium is ACTIVELY able to migrate vertically.

**Response: we will do this.**

Methods:

Line 75: Are three replicate populations enough to assess within species variability? Was this decision based on pilot studies? Were the cultures clonal? Mixed?

**Response: We only used one strain of *Trichodesmium* (IMS101), so intraspecific variability is beyond the scope of our study. Here, population refers to independent replicate culture. We will use other term to eliminate the confusion.**

Line 77: Would be crucial to know where these three temperatures lie on the thermal tolerance/performance curve. The 2007 and 2014 studies just state that these are temperatures that this specific Trichodesmium can live in?

**Response: According to these two papers, we can locate these three temperatures on the thermal tolerance curve which is generally described in line 45-50.**

Line 77: Might have been better to have used a third light intensity toward the Iopt, just for the sake of comparison and to underpin the basic response to temperature of Trichodesmium.

**Response: If Iopt means "optimal light intensity", the high light level in our study is within the range of "optimal light intensity" for this *Trichodesmium* strain. We will clarify this in**

**the revised manuscript.**

Line 77: 160 $\mu$mol quanta m-2 s-1 seem like quite a low light intensity to be saturating, although they report in the supplementary a pilot study that seems confirm the statement. Nevertheless, the cultures for the pilot study were not aerated while it seems to be a constant for Trichodesmium culturing in all other papers (formation of cells$^t$ aggregates and consequently maybe self-shading effects?).

**Response: The value, 160 $\mu$mol quanta m$^{-2}$ s$^{-1}$, is consistent to the values reported and used by other researchers (Garcia et al., 2011; Kranz et al., 2010). Additionally, given the self-shading effects after the formation of cells aggregate, if 160 $\mu$mol quanta m$^{-2}$ s$^{-1}$ is saturating for cultures without aeration, it should also be saturating for cultures with aeration.**

Line 84: 'cyanobacteria were floating singly' consider rephrasing to 'cyanobacteria floated as single filaments'

**Response: We will do this.**

Line 85: Was there a round of pre-acclimation prior to the acclimation phase? Pre-acclimation is a crucial step to avoid carry-over from the previous culture conditions. See for example Trimborn et al 2019, Front. Mar. Sci https://doi.org/10.3389/fmars.2019.00167, Schaum and Collins 2014, Proc Biol Sci.281(1793): 20141486, Scheinin et al 2015 https://doi.org/10.1098/rsif.2015.0056, Lenski 2017 The ISME Journal volume 11, pages2181–2194(2017)

**Response: Yes. All independent cultures were built up from a stock culture which had been kept in 100 $\mu$mol quanta m$^{-2}$ s$^{-1}$ and 25 $^o$C. Subsequently, growth rate of each independent culture was continuously monitored. After the culture was established in the new conditions for 10 generations and its growth rate was stable for more than three consecutive dilutions, we believed that the culture successfully acclimated to**

**the new conditions and started to take samplings. Therefore, carry-over effect should not be a problem here.**

Line 86: How were the growth rate curves fitted? Missing info

**Response: This is described in line 91-94. We will provide more details about how we get the growth rate in the revised manuscript.**

Line 94: should be 'before applying the natural logarithm' instead of ' before natural logarithm'. Generally, how does using Chla as a proxy for growth deal with cells having more Chla per cell?

**Response: Yes, "before applying the natural logarithm" is the correct one.**

**Indeed, Chla:cell ratio was different between cultures grown under different conditions. However, when using Chla as a proxy for growth, what matters is Chla:cell ratio within the culture. For a specific culture, once it acclimates to its growth condition, its Chla:cell ratio is relatively stable. The main variation is the cell cycle-related variation, which can be eliminated by fixing the sampling time and taking samplings during consecutive dilutions (line 91-94). Practically, using Chla as a proxy for growth has also been proven to be a proper method (Breitbarth et al., 2007).**

Line 99: 'acute' as stated above, be mindful of vocabulary used. Define once, then stick to it.

**Response: we will make the corresponding revision to the manuscript.**

Line 102: is 0.5 to read 50 minutes or 30 minutes?  This seems really short for a 25mL vial to equilibrate to the correct temperatures!

**Response: 25 ml was further dispensed into 5 vials (line 100), so it was 5ml-culture that equilibrated to the target temperature in 30 minutes. We had tested this, and it turned out that 30 minutes was enough.**

Line 107:    The Padfield paper is pivotal, but it is not about the Schoolfield equation per se (it is about adaptation to warming and uses the Schoolfield as a tool). The second correct reference is Sharpe, P. J. & DeMichele, D. W. Reaction kinetics of poikilotherm development. J. Theor. Biol. 64, 649–670 (1977).

**Response: The paper mentioned by the referee (Sharpe, P. J. & DeMichele, D. W. Reaction kinetics of poikilotherm development. J. Theor. Biol. 64, 649–670 (1977)) is the origin of the Schoolfield equation, but modifications are proposed to the original equation in (Schoolfield et al., 1981). In our study, the modified Schoolfield equation was used. We will remove the Padfield paper and add the original paper.**

Line 113: Which package was used for the "optimize" function? Which version?

**Response: "optimize" is a built-in function in R language, and the R version is 3.5.3.**

Line 114: If used correctly, the Sharpe-Schoolfield output should not require the 'optimize' function, but simply, rates at Topt can be obtained by re-arranging the equation. It is really not clear at all here how the data were fitted to the Sharpe-    Schoolfield (it clearly went well as the figure looks correct).

**Response: The analytical solution to $T_{opt}$ assumes that $E_a$ is less than $E_h$ (because of the existence of log($1-E_a/E_h$) in the solution). This is satisfied except for one replicate in treatment (31°C, 160 $\mu$mol quanta m$^{-2}$ s$^{-1}$) in our study. Therefore, we resorted to the "optimize" function, which can numerically give the $T_{opt}$.**

Line 114, To me, it would make sense to either extract the parameters (Ea, Eh, Topt. Tc) and

then compare them via a mixed model (e.g. parameter ~ growth temp*light with replicate within treatment as the random effect) or fit a non-linear mixed effects model where lnNrate schoolfield.high(ln.c,Ea,Eh,Th,temp=K,Tc=your Tc value) and, to begin with fixed = list(ln.c + Ea + Eh + Thgrowthtemp*light). You can then compare AICcs of your models (e.g. test also additive effect, each on their own, and just the intercept) and chose the best one. If you compare extracted parameter values, then the MuMin dredge function will come in handy!

**Response: We will re-analyze this part of data using the non-linear mixed effects model. We appreciate the referee's constructive suggestion.**

Line 116: Why was it not possible to measure $N_2$ for samples at 31° C? At what time were the samples taken? I know $N_2$ fixation-related genes show a strong circadian cycle, maybe a similar mechanism is involved?

**Response: We found that the temperature was not homogenous in the multi-zone plant chambers that were used to determine the responses of $N_2$ fixation rate to acute temperature changes, so we used the accurately measured temperatures to do the model fitting. Base on the model, we can get the predicted $N_2$ fixation rate corresponding to the growth temperature.**

**We took the samples in the middle of the light phase, and the circadian rhythm did not play a role here.**

As the authors stated into the nice small "meta-analysis", there is a huge within strains variation, why don't you used more strains? Alternatively, more isolates instead of three if you wanted to assess for within strains variations?

Line 259: What about within-strain variation?

**Response: We interpret the "huge within strains variation" as inter-laboratory variations, which probably comes from the differences in methodological details, such as aeration vs. no-aeration, LED vs. fluorescent lamp etc. However, for a certain study, within strains variation is small. Even with such huge inter-laboratory**

**variations, there is still a trend that light limitation leads to less sensitivity of growth rate to temperature changes in *Trichodesmium* IMS 101. Intra- and inter- strain variations are not the focus of our study.**

Line 129: See comments above – how were the data dealt with? Again, you can either extract parameters and compare via a mixed model, or run a non-linear mixed model starting with the most complex model and then working your way down to the most simple model. For all other phenotypic traits (the ones where you are not fitting a slope), a mixed model seams the way to go!

**Response: We extracted parameters and compared via two-way ANOVA and Turkey test. We will add the paragraphs describing how we analyze the data.**

Results: Throughout: When giving a value, also give the standard deviation or standard error. When referring to the result of statistical test, just giving the post-hoc value is not enough, as that only refers to ONE specific pair-wise comparison. If reporting one specific pair-wise comparison, we need to know which one!

**Response: We will revise our manuscript accordingly, making it traceable.**

Line 140: Might be worth starting out with whether the combined effect of light limitation was indeed interactive, or additive, or if one out of the two described the data best. Without the appropriate reporting of the stats involved, this is impossible to tell.

**Response: We will follow this suggestion.**

Line 141: see above. Strictly speaking, this is not a temperature range, but three temperatures, 23,27,and 31°C.

**Response: We will follow this suggestion.**

Line 145: How much is 'slightly'? Line 152: How much higher is higher? Line 168: What was

the variation around this 1.4 °C increase? Line 183: Add SD or standard error to these values

**Response: We will report the values in the form of mean plus SD or SE, and use more precise vocabularies and specific values to describe our findings in the revised manuscript.**

Line 159: Is acute the same as short-term here? Pick a word, then stick to it.

**Response: Yes. We will follow this suggestion.**

Line 178: be mindful of the tense. Should be 'were able to sustain'

**Response: This will be corrected.**

Line 185: again, not clear what the p value refers to, or what was actually tested in the two way ANOVA

**Response: Similar issues will be properly handled in the revised manuscript.**

Discussion Line 191: "negative growth effects" seems a strong statement, maybe better use "reduced"

**Response: we will follow this suggestion.**

Line 196: level should be levels

**Response: We will correct this.**

Line 202: "temperature is lower" than surface?

**Response: Yes. We will clarify this in the revised manuscript.**

Line 206: maybe I didn't get it, but "respectively" to what?

**Response: These two values were obtained from two papers cited in this sentence. This will be corrected in the revised manuscript.**

Line 210: This is a very nice and clear summary (the additive vs interactive bit), however, without the correct statistical approach it is impossible to tell whether the data actually support this conclusion!

**Response: We will provide this critical information in the revised manuscript.**

Line 232: May need a reference here

**Response: We will add a reference here.**

Line 235: Should be equivalents, not equivalent

**Response: We will correct this.**

Line 250: what is the difference here between 'acclimated" and "short-term"? You mentioned both "short-term temperature norms" and "acclimation" throughout the paper (e.g. Table S1). Please clarify.

**Response: Acclimation is on the scale of multi-generation in which cells have enough time to adjust the gene expression to optimize their performance under current conditions and means that phenotypical traits should be stable. "Short-term" and "acute" are interchangeable, which are on the time scale of several hours (within generation). We will clarify these confusions.**

Line 257: 'a bit different' is too vague Line 258:

**Response: We will replace "a bit different" with "different".**

not sure if 'and/or' is the correct choice of words here.

**Response: We will replace "and/or" with "and".**

Plus, it should be 'on the time scales of acclimation processes'. Consider adding that here,

this is approximately 10 generations.

**Response: We will follow this suggestion.**

Line 266: 'to some extent' is a bit vague, may need a bit more information here.

**Response: We will elaborate this in the revised manuscript.**

Tables Spell out HL and LL as high light and low light You clearly have the data from the light curves in the table, so explaining how you actually got them should not cause too much agony (we hope).

**Response: The data analysis procedures is generally described above. We will explain how we got them in detail in the revised manuscript.**

Figures Might be worth mentioning the software the figures were produced in.

**Response: Software is R (3.5.3) and the package is ggplot2 (3.2.1).**

Figure 1 The lettering of the subpanels as a, b,c, is highly confusing with the significance levels using the same lettering. Might be easier to present the significance levels as a table? What are the slopes in this graph? How were they fitted?

**Response: We will try to present the significance levels as a table.**

**We just linked the near points with lines, so these lines mean nothing special. We will remove the slopes/lines to rule out the confusions.**

Figure 2 Spell out what a.u. stands for. Consider adding confidence intervals to model fits

**Response: a.u. refers to artificial unit. We will add the confidence intervals.**

Figure 4 Not clear where the interactions are. Again, the significance levels are a bit distracting and probably better displayed in a table.

**Response: We will re-analyze this part of data and present the statistical analysis results in a proper way.**

Figure 5: a) Probably good idea to highlight the symbol for this study in bold b) –d) why are there no SDs or confidence intervals?

**Response: a) We will redraw the figure accordingly; b)-d) We will add the intervals.**

Breitbarth, E., Oschlies, A., & LaRoche, J. (2007). Physiological Constraints on the Global Distribution of Trichodesmium - Effect of Temperature on Diazotrophy. Biogeosciences, 4(1), 53-61. doi:10.5194/bg-4-53-2007

Dugdale, R. C., Goering, J. J., & Ryther, J. H. (1964). High Nitrogen Fixation Rates in the Sargasso Sea and the Arabian Sea. Limnology and Oceanography, 9(4), 507-510. doi:10.4319/lo.1964.9.4.0507

Dugdale, R. C., Menzel, D. W., & Ryther, J. H. (1961). Nitrogen Fixation in the Sargasso Sea. Deep Sea Research (1953), 7(4), 297-300. doi:10.1016/0146-6313(61)90051-X

Ehrenberg, C. G. (1830). Neue Beobachlungen Über Blutartige Erscheinungen in Aegypten, Arabien Und Sibirien, Nebst Einer Uebersicht Und Kritik Der Früher Bekannnten. 94(4), 477-514. doi:10.1002/andp.18300940402

Elsner, J. B., Kossin, J. P., & Jagger, T. H. (2008). The Increasing Intensity of the Strongest Tropical Cyclones. Nature, 455, 92-95. doi:10.1038/nature07234

Fu, F. X., Yu, E., Garcia, N. S., et al. (2014). Differing Responses of Marine N2 Fixers to Warming and Consequences for Future Diazotroph Community Structure. Aquatic Microbial Ecology, 72(1), 33-46. doi:10.3354/ame01683

Garcia, N. S., Fu, F. X., Breene, C. L., et al. (2011). Interactive Effects of Irradiance and Co2 on Co2 Fixation and N2 Fixation in the Diazotroph Trichodesmium Erythraeum (Cyanobacteria) 47(6), 1292-1303.

Goering, J. J., Dugdale, R. C., & Menzel, D. W. (1966). Estimates of in Situ Rates of

Nitrogen Uptake by Trichodesmium Sp. In the Tropical Atlantic Ocean. Limnology and Oceanography, 11(4), 614-620. doi:10.4319/lo.1966.11.4.0614

Hong, H., Shen, R., Zhang, F., et al. (2017). The Complex Effects of Ocean Acidification on the Prominent N2 Fixing Cyanobacterium Trichodesmium. Science, 356(6337), 527-531. doi:10.1126/science.aal2981

Hoppe, C. J. M., Flintrop, C. M., & Rost, B. (2018). The Arctic Picoeukaryote Micromonas Pusilla Benefits Synergistically from Warming and Ocean Acidification. Biogeosciences, 15(14), 4353-4365. doi:10.5194/bg-15-4353-2018

Knutson, T. R., McBride, J. L., Chan, J., et al. (2010). Tropical Cyclones and Climate Change. Nature Geoscience, 3, 157. doi:10.1038/ngeo779

Kranz, S. A., Levitan, O., Richter, K.-U., et al. (2010). Combined Effects of Co2 and Light on the N2-Fixing Cyanobacterium Trichodesmium Ims101: Physiological Responses. Plant Physiology, 154(1), 334-345. doi:10.1104/pp.110.159145

Schoolfield, R. M., Sharpe, P. J. H., & Magnuson, C. E. (1981). Non-Linear Regression of Biological Temperature-Dependent Rate Models Based on Absolute Reaction-Rate Theory. Journal of Theoretical Biology, 88(4), 719-731. doi:10.1016/0022-5193(81)90246-0

Trimborn, S., Thoms, S., Karitter, P., et al. (2019). Ocean Acidification and High Irradiance Stimulate the Photo-Physiological Fitness, Growth and Carbon Production of the Antarctic Cryptophyte Geminigera Cryophila. Biogeosciences, 16(15), 2997-3008. doi:10.5194/bg-16-2997-2019

Wehner, M. F., Reed, K. A., Loring, B., et al. (2018). Changes in Tropical Cyclones under Stabilized 1.5 and 2.0 °C Global Warming Scenarios as Simulated by the Community Atmospheric Model under the Happi Protocols. Earth Syst. Dynam., 9(1), 187-195. doi:10.5194/esd-9-187-2018

---

## Author Response (AR1)

**Response to RC1**

The authors investigated the combined effects of light and temperature on the growth, N2 fixation and photosynthesis in the marine diazotroph, Trichodesmium. Light and temperature are two of the most environmental drivers for this species as for other marine primary producers. However, the combined effects of these two factors have surprisingly little been documented on Trichodesmium. This work fills such gap. The new finding from this work is that the thermal responses in Trichodesmium are strongly dependent on light exposures when grown under different light and temp levels. The parameters derived from the measurement are of significance in predicting the re- sponses of Trichodesmium to ocean physical environmental changes associated with global changes. Generally, this work has been well performed and delivers a clear message, but some revisions are needed before being considered acceptable for publication at BG:

1. Line 65, "... where light intensity could be as low as 2 $\mu$mol quanta m-2 s-1". What's the source of this number?

Response: We used the following equation to get this number:

$E(d) = E0 * \exp(-k * d)$

E(d) is the light intensity (μmol quanta m$^{-2}$ s$^{-1}$) at depth d(m); k is the light extinction coefficient; E0 is the surface solar irradiance. We assumed that the water column was homogenous, extinction coefficient was 0.05 m$^{-1}$ (common value reported for subtropical and tropical pelagic oceans (Olson et al., 2015)) and surface solar irradiance was 2000 μmol quanta m$^{-2}$ s$^{-1}$.

In the revised manuscript, we added references at lines 56-57

2. Line 69, "... Trichodesmium's N2 fixation and growth,". It's better to delete " 's N2 fixation and growth".

Response: We revised this paragraph. See lines 71-78

3. Line 115 – 118. In the treatment "light-limiting, 31 oC", the N2 fixation rate under growth condition was obtained through an indirect and unusual way. I recommend that the authors should also take the N2 fixation rate measured at >31oC into consideration (maybe use the average of this and that measured at 30 oC), although such modification may alter the Figure 1b, and require revision of related text.

Response: We re-analyze the short-term thermal response for N$_2$ using nonlinear mixed effects model from which we can directly obtain the N$_2$ fixation rate under growth conditions for all treatments.

4. Line 122. "...Aliquots of 1.5 m ..." should be "1.5 ml".

Response: corrected.

5. The authors should describe the statistical analysis techniques they used in the Material and methods. Although I can roughly deduce the used statistical techniques from the text in Results, the authors should explicitly present them, which will help readers evaluate their results and conclusions.

Response: It was a serious mistake that we omitted the crucial paragraphs describing how we analyzed the data. In the revised manuscript, we added paragraphs at lines 147-164 describing how we analyzed our data.

6. Figure 3. It seems that the selections of temperature gradients are different among different

treatments, which is uncommon. Why? Will this affect the interpretation of the data?

**Response: We found that the temperature was not homogeneous in the multi-zone chambers that were used to measure the response of $N_2$ fixation to short-term temperature changes (Figure 3), so we used the actually measured temperatures rather than the pre-set temperatures.**

**No, this should not be a problem.**

7. Line 202-205. How did the authors get the numbers ">28% and 7%-20%"? The cited literatures do not provide such numbers.

**Response: We got these numbers from the figures in the cited references (Figure 3 in Davis & McGillicuddy, 2006; Fig 8 and 10 in Olson et al., 2015), although they do not show up in the text.**

8. Table 1. In the text, the light treatments were referred as "light limiting" and "light saturating", but in this Table they were denoted as "LL" and "HL". It will be better to keep them consistent.

**Response: Revised accordingly. See Table 2 (Table1 in original manuscript) and Table 3 in the revised manuscript.**

9. Fig 3b. The temperature norm of N2 fixation in the treatment "light-limiting, 31 oC" is quite different from those in other treatments, which deserves more discussion. However, authors didn't put much attention on this phenomenon.

**Response: We guess that the unusual performance in treatment "light-limiting, 31 ºC" might be related to the nitrogenase damage which was induced by the high growth temperature and exacerbated by the light limitation. The quantity of the functional nitrogenase might be not enough to form the expected $N_2$ fixation peak.**

All in all, this work focused on a valuable but previously overlooked scientific topic and obtained some interesting results. If the authors can properly deal with the concerns listed above, I think it will be qualified to be published in BG.

Response to RC2

General comments: This manuscript by Yi et al. examines how light availability (tested at two levels of light intensity) interacts with the effects of warming (along a gradi- ent of three temperatures) in a marine N2 fixer (Trichodesmium erythraeum IMS101) across a time scale of about ten generations. The experiment is in its essence a two- driver question, where either driver might intrinsically decrease or increase metabolic performance, but the

cumulative effect is unknown. The findings and the results are straightforward, with a clearly identifiable general trend. While theoretically relevant (e.g. changes in temperature may coincide with changes in light intensity), it is not quite clear why the authors chose these two drivers over other sets of drivers until much later on in the manuscript. It would also have been nice to see a more explicit evaluation over whether the changes in temperature/light level constitutes an environmental deterioration or amelioration and how that impacts on how they interact. Still, the results are quite interesting, especially since they cover a range of phenotypic traits (growth rates, N2 fixation rates, photosynthetic machinery). However, I have major concerns about how the results are presented: the methods do not indicate how the data were analysed, and the results appear largely as post-hoc output. The latter would indicate that the authors used an ANOVA or similar test, which is indeed indicated more clearly once in line 185, but details are nowhere to be found. For example, a statement about the data is followed up simply by ($p<0.05$, tukey HSD method). It is impossible to glean from this what kind of data were compared and what the original model looked like. As the main question is about interactive effects, and the data are hierarchical in nature (e.g. differently acclimated samples used in a short-term assay), the authors would have needed some kind of mixed model approach. The closest the text ever gets to describing how the data were handled is in line 129 'parameters can be ob- tained through non-linear least squares regression in R language'. Which packages did the authors use to do so? How did they fit their data to the Eiler curve? Similarly, the authors mention the Sharpe-Schoolfield model, but that would be no easy feat with only 5 temperatures (it is a 4 parameter equation). More information would have been crucial here! It clearly worked well, as the fits in Figure 3 don't look too bad. However, we then need to also know how different these curves are from each other. For this, one needs to either extract the parameters and compare them (and describe how!) or run a non-linear mixed effects model (and describe how). As it stands, the handling and analysis of data is not at all traceable. I will provide suggestions on how to deal with this issue in the detailed comments below.

**Response:**

    **We are grateful for the referee's constructive comments and suggestions on our manuscript. We have studied them carefully.**

    **As the referee points out, it would be better if we had explained why we chose light and temperature over other drivers at the beginning of the manuscript. We have revised the Introduction to handle this issue.**

    **It was a serious mistake that we omitted the crucial paragraphs describing how we analyzed the data. We performed the two-way ANOVA with normality (Shapiro-Wilk test) and variance homogeneity (Levene's test) tests to determine whether light, temperature and the interaction of light and temperature affected the phenotypic traits (Figure 1), including growth rate, effective photochemical efficiency and $N_2$ fixation rate. Then, post hoc (Tukey) test was used to do the pairwise comparisons. As with the data in Table 1 and Figure 2, 3, 4, we first extracted the parameters from the non-linear fitting to individual measurement. Then, the two-way ANOVA and Turkey test were used to determine the effects of light, temperature and their interaction on these parameters. The data analysis was done using the R language (version 3.5.3) with the built-in functions, including 'aov', 'shapiro.test' and 'TukeyHSD', function 'nlsLM' from package 'minpack.lm (version 1.2-1)' (line 113) and function 'leveneTest' from package 'car (version 3.0-2)'. We argue that our data analysis processes were appropriate for most of the tested physiological traits.**

**Also, these statistical methods are widely used in other similar work, such as (Hong et al., 2017; Hoppe et al., 2018; Trimborn et al., 2019). Hoverer, we agree with the referee that the part involving the Sharpe-Schoolfield model might be problematic. Using 5 data points to fit a 4-parameter equation was overparameterized. We are grateful that the referee suggests an alternative statistical method to handle this problem, that is, non-linear mixed effects model. We have used this method to re-analyze our data, which did not change our main results and conclusions.**

**In the revised manuscript, we added paragraphs at lines 147-164 describing how we analyzed our data and presented the results in a more traceable way.**

Technical comments and corrections, further suggestions:

Throughout: please double-check use of singular/plural and use of present tense and past tense. Please be careful with the vocabulary used. What is 'acclimation', what is 'short term'? How are either of these different from 'acute'? Be consistent throughout in how you use these words. You could, for example, define them in the introduction and then stick to that definition.

**Response: In our manuscript, "acclimation" means that the cells had been maintained under the growth condition for more than 10 generations with their growth rates being stable. "acute" and "short-term" referred to processes that occur within several hours. We defined these terms at lines 73-76.**

Abstract Line 13: Consider telling the reader which phenotypes from the get go. Line 16: 'range of 23-31' could be misleading, just state the three temperatures Line 16/17: 'when the acclimation … […]… to growth temperature was evaluated by short-term

**Response: We have revised the manuscript accordingly at lines 14-17.**

Line 22: "cells growing under low light levels while distributed deep in the euphotic zone or under cloudy weather conditions might be more susceptible to ocean warm- ing": I would be careful about that, the study refers to response of acclimated cells at different conditions, not to acute or immediate responses (at least for the growth re- sponse), especially when we consider that these cells can actively migrate along the water column.

Line 23: Point out explicitly that this is true for ocean warming occurring on the timescales of a few generations, or, as in your assays, short term responses within the same generation in mere hours. Mention scenarios when this is applicable upfront (mixing, heat waves..)

**Response: We measured such phenotypical traits as growth, $N_2$ fixation, effective quantum yield of *Trichodesmium* cells that had acclimated (over 10 generations) to different light intensity and temperature levels. Additionally, we also measured the response of $N_2$ fixation to short-term (hours) temperature changes. The former is related to the long-term environmental changes, such as global warming, and the latter is more related to strongly disturbed weather conditions, such as cyclones, and marine heat waves. Studies showed that strong cyclones would be more frequent and stronger in the warmer oceans (Elsner et al., 2008; Knutson et al., 2010; Wehner et al., 2018).**

**We have revised these sentences to clarify the ambiguity at lines 21-25.**

Introduction Line 29: might not be all that 'obvious' to all readers. Consider elaborating.

**Response: We have revised this at lines 30-35.**

Line 39: The 1960s are not a century ago yet, plus the literature cited after this statement is pretty recent? Specifically: Is there a reference for the 1960 discovery of diazotrophy in Trichodesmium?

**Response: Modern interest in *Trichodesmium* dates back to the 1960s with the recognition that *Trichodesmium* is diazotrophic.**

**Yes, (Dugdale et al., 1964; Dugdale et al., 1961). We have revised this part and cited these original papers at lines 36-37.**

Line 41/42: 'In the IPCC...[...]' consider rephrasing to 'The IPCC scenario [...] predicts..[..]'

**Response: We have followed this comment at lines 39-40.**

Line 43: I am not sure Collins et al 2013 is the correct reference here, as it is focused on the long-term implications of global climate change, not so much the ocean physics

**Response: The acclimated phenotypic traits, such as growth rate, $N_2$ fixation rate etc., were related to this reference with respect to long term implication. Superimposed on this, we also measured the response of $N_2$ fixation to short-term (hours) temperature change, which was more related to strongly disturbed weather conditions.**

Line 44: 'consequences' on what? Consider elaborating.

**Response: We have revised this at lines 39-44.**

Line 50-52: different responses to warming more due to relation between traits and environment, than only "because of the spatial heterogeneity of present temperatures and projected warming". Clarify it is also a matter of local adaptation.

**Response: Yes. Local adaptation is another factor affecting organism's response to climate change. We have revised and clarified this at lines 45-51 and 76-78.**

Line 68: clearly state that Trichodesmium is ACTIVELY able to migrate vertically.

**Response: revised accordingly at lines 57-58**

Methods:

Line 75: Are three replicate populations enough to assess within species variability? Was this decision based on pilot studies? Were the cultures clonal? Mixed?

**Response: We only used one strain of *Trichodesmium* (IMS101), which was clonal**

**when isolated decades ago, but likely resembles a mixed population now. In our work, the population referred to independent replicate cultures. In the revised manuscript, we used term "cultures" to avoid the confusion. "Three replicate cultures" is widely used in similar studies.**

Line 77: Would be crucial to know where these three temperatures lie on the thermal tolerance/performance curve. The 2007 and 2014 studies just state that these are temperatures that this specific Trichodesmium can live in?

**Response: According to these two papers, we can locate these three temperatures on the thermal tolerance curve. This was described at lines 45-47.**

Line 77: Might have been better to have used a third light intensity toward the Iopt, just for the sake of comparison and to underpin the basic response to temperature of Trichodesmium.

**Response: If Iopt means "optimal light intensity", the high light level in our study is within the range of "optimal light intensity" for this *Trichodesmium* strain. We have clarified this in the revised manuscript at lines 85-86**

Line 77: 160 $\mu$mol quanta m-2 s-1 seem like quite a low light intensity to be saturating, although they report in the supplementary a pilot study that seems confirm the state- ment. Nevertheless, the cultures for the pilot study were not aerated while it seems to be a constant for Trichodesmium culturing in all other papers (formation of cells$^t$ aggregates and consequently maybe self-shading effects?).

**Response: The value, 160 $\mu$mol quanta m$^{-2}$ s$^{-1}$, is consistent with the values reported and used by other researchers (Garcia et al., 2011; Kranz et al., 2010; Cai et al., 2015; Breitbarth et al. 2008). Additionally, given the self-shading effects after the formation of cells aggregate, if 160 $\mu$mol quanta m$^{-2}$ s$^{-1}$ is saturating for cultures without aeration, it should also be saturating for cultures with aeration.**

Line 84: 'cyanobacteria were floating singly' consider rephrasing to 'cyanobacteria floated as single filaments'

**Response: revised accordingly at line 91**

Line 85: Was there a round of pre-acclimation prior to the acclimation phase? Pre-acclimation is a crucial step to avoid carry-over from the previous culture conditions. See for example Trimborn et al 2019, Front. Mar. Sci https://doi.org/10.3389/fmars.2019.00167, Schaum and Collins 2014, Proc Biol Sci.281(1793): 20141486, Scheinin et al 2015 https://doi.org/10.1098/rsif.2015.0056, Lenski 2017 The ISME Journal volume 11, pages2181–2194(2017)

**Response: Yes. All independent cultures were built up from a stock culture which had been kept in 100 $\mu$mol quanta m$^{-2}$ s$^{-1}$ and 25 ºC. Subsequently, growth rate of each independent culture was continuously monitored. After the culture was established in the new conditions for 10 generations and its growth rate was stable for more than three consecutive dilutions, we believed that the culture adequately acclimated to the new conditions and started to take samples (see lines 91-93 in the revised manuscript).**

**Therefore, carry-over effect should not be a problem here.**

Line 86: How were the growth rate curves fitted? Missing info

**Response: In the original manuscript, this was described in lines 91-94. We provided more details about how we obtained the growth rate in the revised manuscript at lines 100-105 and 148-149.**

Line 94: should be 'before applying the natural logarithm' instead of ' before natural logarithm'. Generally, how does using Chla as a proxy for growth deal with cells having more Chla per cell?

**Response: Indeed, Chla:cell ratio was different between cultures grown under different conditions. However, when using Chla as a proxy for growth, what matters is Chla:cell ratio within the culture. For a specific culture, once it acclimates to its growth condition, its Chla:cell ratio is relatively stable. The main variation is the cell cycle-related variation, which can be eliminated by fixing the sampling time and taking samplings during consecutive dilutions. Practically, using Chla as a proxy for growth has also been proven to be a proper method (Breitbarth et al., 2007).**

Line 99: 'acute' as stated above, be mindful of vocabulary used. Define once, then stick to it.

**Response: we made the corresponding revisions to the manuscript as mentioned above.**

Line 102: is 0.5 to read 50 minutes or 30 minutes? This seems really short for a 25mL vial to equilibrate to the correct temperatures!

**Response: 25 ml was further dispensed into 5 vials, so it was 5ml-culture that equilibrated to the target temperature in 30 minutes (line 110-111 in the revised manuscript). We had tested this, and it turned out that 30 minutes was enough.**

Line 107: The Padfield paper is pivotal, but it is not about the Schoolfield equation per se (it is about adaptation to warming and uses the Schoolfield as a tool).The second correct reference is Sharpe, P. J. & DeMichele, D. W. Reaction kinetics of poikilotherm development. J. Theor. Biol. 64, 649–670 (1977).

**Response: The paper mentioned by the referee (Sharpe, P. J. & DeMichele, D. W. Reaction kinetics of poikilotherm development. J. Theor. Biol. 64, 649–670, 1977) is the origin of the Schoolfield equation, but modifications have been made by Schoolfield et al. (1981) and Padfield et al. 2015. In our study, the modified Schoolfield equation was used. We add the original paper at lines 117-118.**

Line 113: Which package was used for the "optimize" function? Which version?

Line 114: If used correctly, the Sharpe-Schoolfield output should not require the 'optimize' function, but simply, rates at Topt can be obtained by re-arranging the equation. It is really not clear at all here how the data were fitted to the Sharpe- Schoolfield (it clearly went well as the figure looks correct). To me, it would make sense to either extract the parameters (Ea, Eh, Topt. Tc) and then compare them via a mixed model (e.g. parameter growth ~

temp*light with replicate within treatment as the random effect) or fit a non-linear mixed effects model where lnNrate schoolfield.high(ln.c,Ea,Eh,Th,temp=K,Tc=your Tc value) and, to begin with fixed = list(ln.c + Ea + Eh + Th growthtemp*light). You can then compare AICcs of your mod- els (e.g. test also additive effect, each on their own, and just the intercept) and chose the best one. If you compare extracted parameter values, then the MuMin dredge function will come in handy!

**Response: "optimize" is a function in package "stats" in R language.**

**The analytical solution to $T_{opt}$ given by Padfield et al. 2015 assumes that $E_a$ is less than $E_h$ (because of the existence of $\log(1-E_a/E_h)$ in the solution), which was not always satisfied in our original data analysis. However, we just found that this analytical solution was incorrect, and gave the correct one in the revised manuscript at lines 127. In the revised manuscript, with the correct analytical solution to $T_{opt}$, we have used the nonlinear mixed effects model to re-analyze the short-term thermal response for $N_2$ fixation (lines 155-160). We appreciate the referee's constructive suggestion.**

Line 116: Why was it not possible to measure N2for samples at 31° C? At what time were the samples taken? I know N2 fixation-related genes show a strong circadian cycle, maybe a similar mechanism is involved?

**Response: We found that the temperature was not homogenous in the multi-zone plant chambers that were used to determine the responses of $N_2$ fixation rate to acute temperature changes, so we used the accurately measured temperatures (which did not cover 31 ºC) to do the model fitting. Base on the model, we were able to get the predicted $N_2$ fixation rates corresponding to the growth temperature.**

**We took all the samples in the middle of the light phase for all the treatments, and the circadian rhythm did not play a role here.**

As the authors stated into the nice small "meta-analysis", there is a huge within strains variation, why don$^t$t you used more strains? Alternatively, more isolates instead of three if you wanted to assess for within strains variations?

**Response: Almost all laboratory studies exploring the effects of temperature on *Trichodesmium* use the strain IMS 101, so we are not able to use more strains. We interpret the "huge within strains variation" as inter-laboratory variations, which probably comes from the differences in methodological details, such as aeration vs. no-aeration, LED vs. fluorescent lamp etc. However, for a certain study, variations within strains are small. Even with such huge inter-laboratory variations, there is still a trend that light limitation leads to less sensitivity of growth rate to temperature changes in *Trichodesmium* IMS 101.**

Line 129: See comments above – how were the data dealt with? Again, you can either extract parameters and compare via a mixed model, or run a non-linear mixed model starting with the most complex model and then working your way down to the most simple model. For all other phenotypic traits (the ones where you are not fitting a slope), a mixed model

seams the way to go!

**Response: We extracted parameters and compared via two-way ANOVA and Turkey test. We added the paragraphs describing how we analyzed these data at lines 161-164.**

Results: Throughout: When giving a value, also give the standard deviation or standard error. When referring to the result of statistical test, just giving the post-hoc value is not enough, as that only refers to ONE specific pair-wise comparison. If reporting one specific pair-wise comparison, we need to know which one!

**Response: In the revised manuscript, "Results" have been revised accordingly, making it traceable.**

Line 140: Might be worth starting out with whether the combined effect of light limita- tion was indeed interactive, or additive, or if one out of the two described the data best. Without the appropriate reporting of the stats involved, this is impossible to tell.

Line 141: see above. Strictly speaking, this is not a temperature range, but three tempera- tures, 23,27,and 31oC.

**Response: We followed this suggestion and revised the text at lines 167-180**

Line 145: How much is 'slightly'? Line 152: How much higher is higher? Line 168: What was the variation around this 1.4 oC increase? Line 183: Add SD or standard error to these values

**Response: We have reported the values in the form of mean plus SD or SE, and used more precise vocabularies and specific values to describe our findings in the revised manuscript at lines178-180, 209-211 and 228-230.**

Line 159: Is acute the same as short-term here? Pick a word, then stick to it.

**Response: Yes. We followed this suggestion and revised throughout our text.**

Line 178: be mindful of the tense. Should be 'were able to sustain'

**Response: Revised.**

Line 185: again, not clear what the p value refers to, or what was actually tested in the two way ANOVA

**Response: Revised accordingly throughout our text.**

Discussion Line 191: "negative growth effects" seems a strong statement, maybe better use "reduced"

**Response: Revised accordingly at lines 233-234.**

Line 196: level should be levels

**Response: Corrected.**

Line 202: "temperature is lower" than surface?

**Response: Yes. We have clarified this in the revised manuscript at line 247.**

Line 206: maybe I didn't get it, but "respectively" to what?

**Response: These two values were obtained from two papers cited in this sentence. We have corrected this in the revised manuscript at lines 250-252.**

Line 210: This is a very nice and clear summary (the additive vs interactive bit), how- ever, without the correct statistical approach it is impossible to tell whether the data ac- tually support this conclusion!

**Response: We provided this critical information in the revised manuscript at lines 147-164.**

Line 232: May need a reference here

**Response: A reference was added at lines 276.**

Line 235: Should be equivalents, not equivalent

**Response: Corrected.**

Line 250: what is the difference here between 'acclimated" and "short-term"? Youmentioned both "short-term temperature norms" and "acclimation" throughout the pa- per (e.g. Table S1). Please clarify.

**Response: We apologize for having used the confusing wordings. Now, we have clarified this, as indicated in the above response, at lines 73-76**

Line 257: 'a bit different' is too vague Line 258: not sure if 'and/or' is the correct choice of words here. Plus, it should be 'on the time scales of acclimation processes' . Consider adding that here, this is approximately 10 generations. Line 259: What about within-strain variation?

**Response: We revised these at lines 295-305.**

Line 266: 'to some extent' is a bit vague, may need a bit more informationhere.

**Response: We have revised this part at lines 310-315**

Tables Spell out HL and LL as high light and low light

**Response: Revised accordingly.**

You clearly have the data from the light curves in the table, so explaining how you actually got them should not cause too much agony (we hope).

**Response: We explained how we got them in detail in the revised manuscript at lines 147-164.**

Figures Might be worth mentioning the software the figures were produced in.

**Response: Software is R (3.5.3) and the packages are ggplot2 (3.2.1) and plot3D(1.1.1).**

Figure 1 The lettering of the subpanels as a, b,c, is highly confusing with the signifi- cance levels using the same lettering. Might be easier to present the significance levels as a table? What are the slopes in this graph? How were they fitted?

**Response: Revised accordingly.**

**We just linked the near points with lines, so these lines mean nothing special. We removed the slopes/lines to rule out the confusions.**

Figure 2 Spell out what a.u. stands for. Consider adding confidence intervals to model fits

**Response: a.u. refers to artificial unit. We added the confidence intervals in Figure 2.**

Figure 4 Not clear where the interactions are. Again, the significance levels are a bit distracting and probably better displayed in a table.

**Response: The information provided in original Figure 4 was presented in Table 3 in revised manuscript**

Figure 5: a) Probably good idea to highlight the symbol for this study in bold b) –d) why are there no SDs or confidence intervals?

**Response: We redrew the original "Figure 5 panel a" accordingly as Figure 4 in the revised manuscript. Original "Fig 5 panel b-d" were removed for reasons (see lines 295-304).**

**References cited in the above responses**

Breitbarth, E., Oschlies, A., & LaRoche, J. (2007). Physiological Constraints on the Global Distribution of Trichodesmium - Effect of Temperature on Diazotrophy. Biogeosciences, 4(1), 53-61. doi:10.5194/bg-4-53-2007

Dugdale, R. C., Goering, J. J., & Ryther, J. H. (1964). High Nitrogen Fixation Rates in the Sargasso Sea and the Arabian Sea. Limnology and Oceanography, 9(4), 507-510. doi:10.4319/lo.1964.9.4.0507

Dugdale, R. C., Menzel, D. W., & Ryther, J. H. (1961). Nitrogen Fixation in the Sargasso Sea. Deep Sea Research (1953), 7(4), 297-300. doi:10.1016/0146-6313(61)90051-X

Elsner, J. B., Kossin, J. P., & Jagger, T. H. (2008). The Increasing Intensity of the Strongest Tropical Cyclones. Nature, 455, 92-95. doi:10.1038/nature07234

Fu, F. X., Yu, E., Garcia, N. S., et al. (2014). Differing Responses of Marine N2 Fixers to Warming and Consequences for Future Diazotroph Community Structure. Aquatic Microbial Ecology, 72(1), 33-46. doi:10.3354/ame01683

Garcia, N. S., Fu, F. X., Breene, C. L., et al. (2011). Interactive Effects of Irradiance and $CO_2$ on $CO_2$ Fixation and N2 Fixation in the Diazotroph Trichodesmium Erythraeum (Cyanobacteria) 47(6), 1292-1303.

Hong, H., Shen, R., Zhang, F., et al. (2017). The Complex Effects of Ocean Acidification on the Prominent N2 Fixing Cyanobacterium Trichodesmium. Science, 356(6337), 527-531. doi:10.1126/science.aal2981

Hoppe, C. J. M., Flintrop, C. M., & Rost, B. (2018). The Arctic Picoeukaryote Micromonas Pusilla Benefits Synergistically from Warming and Ocean Acidification. Biogeosciences, 15(14), 4353-4365. doi:10.5194/bg-15-4353-2018

Knutson, T. R., McBride, J. L., Chan, J., et al. (2010). Tropical Cyclones and Climate Change. Nature Geoscience, 3, 157. doi:10.1038/ngeo779

Kranz, S. A., Levitan, O., Richter, K.-U., et al. (2010). Combined Effects of Co2 and Light on the N2-Fixing Cyanobacterium Trichodesmium Ims101: Physiological Responses. Plant Physiology, 154(1), 334-345. doi:10.1104/pp.110.159145

Schoolfield, R. M., Sharpe, P. J. H., & Magnuson, C. E. (1981). Non-Linear Regression of Biological Temperature-Dependent Rate Models Based on Absolute Reaction-Rate Theory. Journal of Theoretical Biology, 88(4), 719-731. doi:10.1016/0022-5193(81)90246-0

Trimborn, S., Thoms, S., Karitter, P., et al. (2019). Ocean Acidification and High Irradiance Stimulate the Photo-Physiological Fitness, Growth and Carbon Production of the Antarctic Cryptophyte Geminigera Cryophila. Biogeosciences, 16(15), 2997-3008. doi:10.5194/bg-16-2997-2019

[revised manuscript text omitted]

